# An emulator-based modelling framework for studying astronomical controls on ocean anoxia with an application to the Devonian

**Loïc Sablon[1], Pierre Maffre[2,3], Yves Goddéris[2], Paul J. Valdes[4], Justin Gérard[1], Jarno J. C. Huygh[5], Anne-Christine Da Silva[5], and Michel Crucifix[1]**

[1]Earth and Life Institute, UCLouvain, Louvain-la-Neuve, Belgium
[2]Géosciences Environnement Toulouse, CNRS – Université Paul Sabatier, Toulouse, France
[3]Aix-Marseille Université, CNRS, IRD, INRA, Coll. France, CEREGE, Aix-en-Provence, France
[4]School of Geographical Sciences, University of Bristol, Bristol, BS8 1SS, United Kingdom
[5]SediCClim, ULiège, Liège, Belgium

**Correspondence:** Loïc Sablon (loic.sablon@uclouvain.be)

**Abstract.** We present a modelling framework to study the response of continental flux dynamics and ocean anoxia to astronomical forcing. The GEOCLIM model is coupled with a Gaussian Process-based climate emulator, designed to efficiently capture the global distribution of temperature and precipitation as simulated by the general circulation model HadCM3. The emulator employs principal component analysis for dimensionality reduction. Compared to earlier approaches, our emulator features an additive kernel function that better captures the spatial complexity of ocean responses and accounts for ocean heat transport. This setup facilitates interactive coupling between $CO_2$ levels through an iterative procedure involving GEOCLIM and the emulator, enabling systematic exploration of various orbital and $pCO_2$ configurations. We demonstrate the model's capabilities with a Devonian case study, revealing an emergent relationship between high eccentricity periods and enhanced regolith development, though with limited impact on phosphorus fluxes to the ocean.

## 1 Introduction

Modelling long-term climate dynamics and related environmental phenomena presents significant challenges, particularly when investigating events from deep geological time. These challenges stem from uncertainties in boundary conditions, forcing mechanisms, and geological reconstructions, which complicate parameter tuning, as well as from the long timescales involved. A fundamental computational bottleneck emerges when attempting to couple detailed climate modelling with biogeochemical processes over geological timescales: while spatially resolved climate information is essential for understanding continental weathering and nutrient transport, the computational demands of General Circulation Models (GCMs) make multimillion-year integrations impractical.

The GEOCLIM framework (Donnadieu et al., 2006; Arndt et al., 2011; Goddéris and Donnadieu, 2019; Maffre et al., 2021) has been developed as a versatile suite of models designed to dynamically simulate geochemical cycles and regolith (i.e., the upper layer of the continental crust where chemical weathering occurs) dynamics over long timeframes. Its computational efficiency stems from a box model approach for ocean biogeochemistry, combined with a spatially resolved continental module that captures weathering and erosion processes. However, GEOCLIM faces a critical limitation: realistically estimating the response of atmospheric conditions, particularly the impact of astronomical forcing on rainfall patterns that control nutrient mobilization, requires detailed climate inputs that are computationally prohibitive to generate over geological timescales using traditional GCM approaches.

Climate emulation provides an elegant solution to this incompatibility. By training statistical models on carefully designed ensembles of GCM experiments, emulators can cap-

ture the spatial complexity of the ocean-atmosphere response to astronomical forcing as simulated by the GCM, while requiring minimal computational resources once trained. This efficiency enables systematic exploration of multiple astronomical scenarios and their impact on nutrient fluxes and biogeochemical cycles, bridging the gap between detailed climate modelling and long-term Earth system investigations.

In this work, we advance the emulation paradigm by developing a dynamically coupled system that introduces several key methodological innovations. Building upon the framework proposed by Bounceur et al. (2015), Lord et al. (2017) and Van Breedam et al. (2021) and the theory of Gaussian Processes (GPs, MacKay, 1998; Rasmussen, 2006), our approach features an additive kernel function that better captures spatial complexity in climate responses, a two-tier approach using both coupled and slab ocean models to account for ocean heat transport, and full bidirectional coupling that enables interactive feedback between atmospheric $CO_2$ levels and climate patterns. The geochemical model determines $CO_2$ levels, which the emulator then translates into climate predictions affecting continental weathering and erosion, and consequently nutrient fluxes. These, in turn, influence subsequent geochemical calculations, allowing for an iterative feedback process that can reveal how astronomical forcing ultimately impacts continental fluxes and ocean biogeochemistry.

The astronomical forcing of interest involves changes in the rotation of the Earth's axis and the geometry of its orbit, which impact seasonal variations and the level of incoming solar radiation, thereby affecting global climate (see Berger and Yin, 2021 or Zeebe and Kocken, 2024 for an introduction on astronomical forcing). These orbital cycles operate on timescales ranging from tens of thousands to millions of years, creating long-term modulations in climate patterns that can influence continental weathering, nutrient storage and release, and ultimately ocean biogeochemistry. Understanding these connections requires modelling approaches capable of capturing both the spatial complexity of climate responses and the temporal scales of biogeochemical processes.

To validate our methodological advances, we demonstrate the framework's capabilities with an application to the Devonian period (419 to 359 million years ago). The Devonian serves as an excellent test case, as it was characterised by significant climatic and environmental transformations, including 29 instances of ocean anoxic or hypoxic conditions (Becker et al., 2020). Climate modelling studies have shown this period to be sensitive to changes in atmospheric $CO_2$ concentrations and orbital configuration (Brugger et al., 2019), making it ideal for testing our emulator framework. It has been hypothesized that astronomical cycles may influence the recurrence of anoxia through their effects on regional climate patterns, particularly tropical precipitation, which controls nutrient mobilization and ocean biogeochemistry (De Vleeschouwer et al., 2017; Wichern et al., 2024). Testing these hypotheses requires precisely the type of spatially and temporally resolved modelling that our emulator-GEOCLIM coupling provides.

The primary objective of this paper is to provide a comprehensive description of our emulator-based modelling framework, along with the justification for our technical choices. This serves as a methodological foundation for investigating how astronomical forcing impacts continental fluxes and ocean biogeochemistry across geological timescales. It is essential to ensure that the GP emulator is robust enough for integration into a coupled soil-biogeochemical model. Though we illustrate its application with the Devonian case study, the methodology remains applicable to other geological periods and research questions requiring long-term Earth system modelling.

## 2 Methods

Following the definition by Kennedy and O'Hagan (2000), Oakley and O'Hagan (2002) and Wilkinson (2010), an emulator is a statistical model that predicts the output of climate conditions provided by a "simulator", here a GCM, in response to input parameters. The emulator must first be trained with a set of experiments with the GCM; the experiment set is designed such as to efficiently probe an input parameter space. Once trained, the emulator enables swift approximations of climate outputs, thereby allowing us to explore scenarios that would be practically impossible to produce through a traditional GCM usage.

To construct the climate database used to train the emulator, we decided to employ two types of GCMs: a coupled ocean-atmosphere model and a lighter, computationally less expensive model where the ocean is represented by a flat layer called a slab (we hereafter call it the slab model). We follow a two-tier approach by which we calibrate the slab model with the coupled one, and then produce a larger set of experiments with the slab model to generate the data needed to train the emulator. This double-layer strategy enables us to capture essential atmospheric dynamics accounting for changes in ocean heat transport while mitigating the computational cost. We now give the technical details of this procedure.

### 2.1 Geophysical models setup and boundary conditions

#### 2.1.1 HadCM3L and HadSM3

The Hadley Centre Coupled Model version 3 Low Resolution (HadCM3L) serves as our primary reference GCM for simulating Devonian climate conditions. The spatial resolution of 3.75° longitude × 2.5° latitude for both the atmosphere and oceanic grid can adequately capture the essential mechanics of monsoonal systems. Albedo, snow accumulation, vegetation and soil processes were computed using the MOSES-

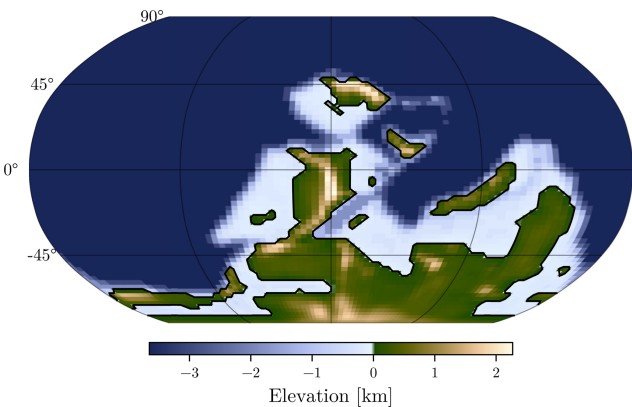

**Figure 1.** Topography of the Late Devonian (370 Ma) according to Scotese and Wright (2018) palaeo-digital elevation model, and downscaled to match the grid resolution of HadCM3L.

II land-surface scheme (Cox et al., 1999, 2000; Cox, 2001; Essery et al., 2001). For a comprehensive description of the model's technical aspects, the reader is directed to its descriptive paper (Valdes et al., 2017).

HadSM3 is the slab version of HadCM3 where the atmosphere is coupled to a simple non-dynamic mixed layer ocean (Williams et al., 2005). This simpler model is computationally efficient in two ways: ocean dynamics are by-passed, and a stationary solution can be found after an integration
of ca. 30 years, compared to 5000+ years with HadCM3 (Valdes et al., 2021). Hence, it will be used here to perform a larger number of experiments.

The model uses large input files called *dumps*, which contain initial and boundary conditions. Those used here are
adapted from Valdes et al. (2021). The 370 Ma palaeogeography is obtained from the dataset of Scotese and Wright (2018), as depicted in Fig. 1. It was chosen with the intention to best represent the Earth's geometry during the Frasnian-Famennian boundary, coinciding with an OAE that triggered
a mass extinction around 372 million years ago (Percival et al., 2018; Carmichael et al., 2019).

Given the slab ocean model's deficiency in ocean dynamics, the computation of a corrective flux, known as the Anomalous Heat Convergence (AHC), is required to accu-
rately represent sea-surface temperatures (SSTs) and sea-ice concentration.

### 2.1.2 Anomalous heat convergence

In a slab model, the atmosphere is coupled to a much simpler resolution of the ocean, having a single layer. This coupling
has the advantage of still capturing the first order interaction of the ocean without the need of running experiments for several thousands of years, as the response is much faster. However, the model must be calibrated on realistic ocean to be able to capture its dynamic.

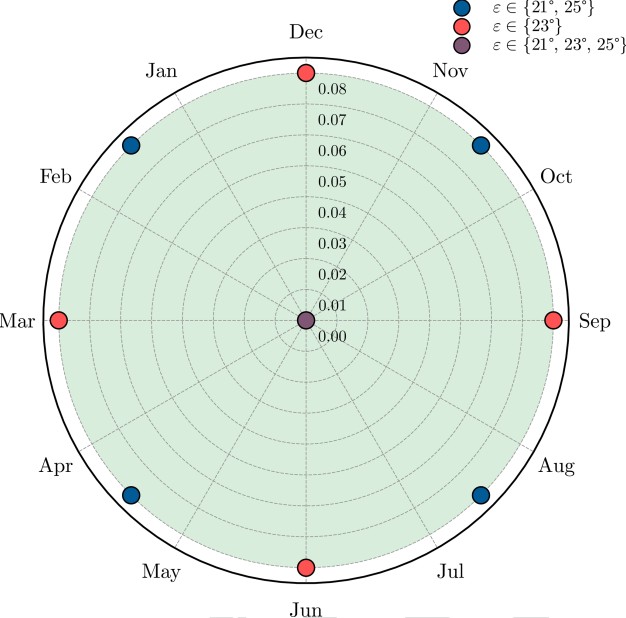

**Figure 2.** Slices of the cylinder constituting our HadCM3 [TS1] experimental plan, with the eccentricity as the radius and the longitude of perihelion as the azimuthal angle. The angle is expressed in the month when the Earth is closest to the Sun, such that the vertical axis is the climatic precession $e \sin \varpi$ and the horizontal axis is the coprecession $e \cos \varpi$. The colour is associated to obliquity $\varepsilon$ being the third dimension. The green area illustrates the possible values used during the interpolation process.

To transition from the coupled model (HadCM3L) to the
slab model (HadSM3), we follow a two-step approach. First, a series of 15 experiments were conducted using the coupled version of the model with varying orbital parameters (Fig. 2). These experiments were run for 250 years, sufficient to achieve a near-equilibrium state in the upper ocean. For
each experiment, the resulting SST and sea-ice concentration fields from HadCM3L were then prescribed to the slab model to diagnose the AHC. This heat convergence field represents the heat transport that would have been provided by ocean dynamics in the coupled model, allowing the slab model to
replicate similar climate patterns despite its simplified ocean representation. Second, we use an Inverse Distance Weighting interpolation (IDW) to obtain appropriate heat convergence fields for our full set of 81 experiments with different orbital configurations (cf. Fig. 4).

The [TS2] 15 experiments using HadCM3 were designed with varying orbital parameters (see Fig. 4). After 250 years, the upper layer of the ocean was close to equilibrium, and the simulation was stopped (the deep ocean could take more than 3000 years of model run). The obtained SST and sea-ice
concentration fields were then prescribed to the slab model to diagnose the AHC. At steady state, the temporal and spatial mean of the field should be equal to $0\,\mathrm{W\,m^{-2}}$. A non-zero value is equivalent to applying a constant forcing to the

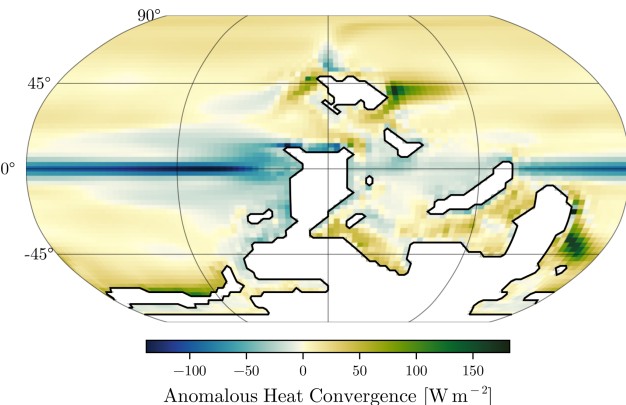

**Figure 3.** Ensemble mean of the annual Anomalous Heat Convergence, expressed in $W\,m^{-2}$.

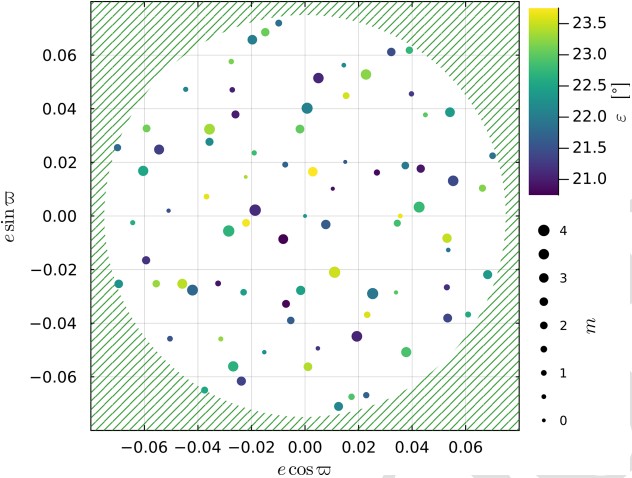

**Figure 4.** Projection of the experimental design onto the $\{e\sin\varpi, e\cos\varpi\}$ plane. Each point represents a GCM experiment. The colour is associated to obliquity $\varepsilon$ (in degrees) and the marker size to carbon dioxide $m = \log_2\left(\frac{pCO_2}{280\,\text{ppm}}\right)$. The dashed area is outside the range of our maximal eccentricity $e_{\max} = 0.075$.

model. It appears that our set of experiments exhibits this non-desired behaviour, in part coming from the fact that the whole system was not exactly at equilibrium after the time period. In order to remove that parameter-dependent bias, a constant value is added to each such that the resulting average forcing is the same for each experiment, with a value of $-0.486\,W\,m^{-2}$. Figure 3 illustrates the average of the annual AHC across the set of 15 experiments.

Each of our 81 HadSM3 experiments should have a AHC field prescribed that is consistent with the information obtained from the 15 HadCM3 experiments. With 15 experiments only, it is more parsimonious and practical to estimate AHC using IDW interpolation, rather than with a more complex Gaussian Process Regression.

We designate $\boldsymbol{x}_i = [\tilde{e}_i \sin\varpi_i, \tilde{e}_i \cos\varpi_i, \tilde{\varepsilon}_i]^\top \in [-1, 1]^3$ as the vector that encapsulates the orbital parameters. These parameters are linearly rescaled to ensure that each one falls within the same range.

Subsequently, an interpolated value of the AHC can be computed as the weighted sum of the entire set. In this context, each weight is determined by the inverse square distance between the new point and the data, indicating that the weight is inversely proportional to the square of the distance.

$$\mathbf{Y}(\boldsymbol{x}) = \begin{cases} \mathbf{Y}_i & \text{if } \boldsymbol{x} = \boldsymbol{x}_i \\ \dfrac{\sum\limits_{i=1}^{N} (\boldsymbol{x}-\boldsymbol{x}_i)^{-2}\mathbf{Y}_i}{\sum\limits_{k=1}^{N} (\boldsymbol{x}-\boldsymbol{x}_k)^{-2}} & \text{otherwise} \end{cases} \tag{1}$$

### 2.1.3 Simulation setup

For all simulations, the palaeogeography and the solar constant ($1322\,W\,m^{-2}$, Bahcall et al., 2001) are kept identical. The orbital parameters (eccentricity, longitude of the perihelion, and obliquity), the atmospheric $CO_2$ concentration and the monthly heat-correction flux are constant in time within a given experiment but prescribed to different values across experiments. Once these boundary conditions and parameters are established, HadSM3 is run for 80 model years, with the last 30 years averaged to minimise interannual variability and accurately represent annual climatic averages of key variables, such as temperature, evaporation, and precipitation.

### 2.1.4 GEOCLIM

GEOCLIM (Donnadieu et al., 2006; Arndt et al., 2011; Goddéris and Donnadieu, 2019; Maffre et al., 2021) is a versatile suite of models designed to dynamically simulate geochemical cycles over geological timescales. The core of the model structure is COMBINE, an ocean-atmosphere chemistry model based on advection-reaction principles, using a discretisation into 10 boxes representing nine oceanic and one atmospheric compartments. More precisely, we use version 6.1.0, which differs slightly from its newest version GEOCLIM7 (Maffre et al., 2025). However, the continental modules of both versions are the same, which is the key point of this study. The coupling with the emulator developed in this work creates the integrated framework we designate as `GEOCLIM6.1-EMUL`.

The nine oceanic boxes represent the following spatial and depth discretisation: polar deep and surface (including thermocline) for both northern and southern hemispheres, mid-latitude deep, intermediate (thermocline), and surface, and epicontinental deep and surface (receiving the continental fluxes). This box structure enables the representation of the main oceanic water masses and their biogeochemical characteristics whilst maintaining computational efficiency for long-term simulations.

This model is integrated with an early diagenesis module responsible for calculating output fluxes, such as the burial of elements in marine sediments across each box. Additionally, GEOCLIM features a continental module, which computes input fluxes and is spatially resolved using a geographic mesh grid. For this study, the continental module is configured to use the same Late Devonian paleogeography (370 Ma) as employed in the HadCM3/HadSM3 simulations, ensuring consistency between the climate emulator and biogeochemical model. This module is adaptable and includes a soil model based on previous work by Gabet and Mudd (2009) and West (2012).

GEOCLIM computes various geochemical cycles, including those of carbon, oxygen, alkalinity, phosphorus, and sulfur, within each reservoir. It dynamically simulates these cycles without assuming steady-state conditions, incorporating rapid processes like ocean mixing and water column sedimentation under parametrised forms. However, it necessitates inputs from a climate model.

Within the continental module of GEOCLIM, DynSoil accounts for physical erosion and silicate weathering. The other fluxes computed by the continental module are carbonate weathering, petrogenic organic carbon weathering, phosphorus weathering, and terrestrial organic carbon export. Surface bedrock is categorized into six lithological classes, following Park et al. (2020), utilising the data compilation of Hartmann and Moosdorf (2012). The erosion and silicate weathering components solve the governing equations dynamically instead of assuming a steady-state regolith, enabling the model to capture transient responses to climate perturbations and the coupling between physical erosion and chemical weathering rates. Regarding petrogenic organic carbon and sulfide weathering, these fluxes are considered proportional to the modelled erosion rate with prescribed organic matter content.

A key aspect of DynSoil is that it resolves the regolith thickness and computes the weathering fluxes accordingly. Here, we call *regolith* the interface between unweathered bedrock and the Earth's surface where chemical weathering reactions occur. It can be thought of as the layer of unconsolidated material that covers the bedrock. The regolith model describes a vertical profile of the abundance of primary minerals, beginning with an abundance of 1 at the regolith/bedrock transition and decaying towards the surface as a result of dissolution.

In GEOCLIM6.1.0, phosphorus weathering represents a key biogeochemical flux, as being the only nutrient represented, delivered to the ocean proportionally through three distinct weathering sources: silicate, carbonate, and petrogenic organic carbon. Critically, only silicate weathering is dynamically computed within DynSoil and therefore directly affected by regolith thickness variations. Carbonate weathering and petrogenic organic carbon weathering contribute to phosphorus delivery but are computed independently of regolith dynamics. This coupling between regolith thickness and silicate weathering explains why changes in regolith dynamics may not always translate directly to proportional changes in total phosphorus fluxes, as the regolith-dependent component represents only one of the three phosphorus sources.

The DynSoil module needs inputs of temperature and runoff (or precipitation and evaporation) over a spatial grid, which are typically supplied by a GCM or, here, the emulator of the GCM.

In summary, assuming adequate climate forcing, GEOCLIM's integration of continental processes, climate dynamics, and ocean biogeochemistry enables comprehensive investigation of Earth system dynamics over multimillion-year timescales as needed for our objective.

## 2.2 Experimental design

### 2.2.1 Parameter space

The inner solar system's dynamics exhibit chaotic behaviour, making it impossible to pinpoint an exact astronomical solution for the Late Devonian period. Only a handful of constraints on cycle frequencies, stabilities and amplitudes remain (Laskar et al., 2004, 2011; Mogavero and Laskar, 2022; Zeebe and Kocken, 2024; Zeebe and Lantink, 2024).

Since the objective of this study is to construct climatologies for a plausible Late Devonian orbital configuration, it requires an extensive exploration of a broad domain. We focus on the response to four parameters: eccentricity $e$ (which characterises the shape of the Earth's orbit), longitude of the perihelion $\varpi$ (the angular distance between the September equinox and the point in Earth's orbit where it is closest to the Sun – heliocentric convention), obliquity $\varepsilon$ (the tilt of the ecliptic relative to the celestial equator), and the partial pressure of carbon dioxide in the atmosphere $p\mathrm{CO}_2$.

The following intervals for each parameter were chosen to make sure that no extrapolation outside the domain will be needed:

$$e \in [0, 0.075] \tag{2}$$

$$\varpi \in [0, 2\pi[ \tag{3}$$

$$\varepsilon \in [20.75°, 23.75°] \tag{4}$$

$$p\mathrm{CO}_2 \in [1, 16] \times 280\,\mathrm{ppm}, \tag{5}$$

We propose the hypothesis that the prior basis $\{e\sin\varpi, e\cos\varpi, \varepsilon, \log_2(p\mathrm{CO}_2)\}$, where $e\sin\varpi$ denotes the climatic precession and $p\mathrm{CO}_2$ is expressed relative to pre-industrial levels (i.e., in units of 280 ppm), optimally encodes the initial distance metric among different climatologies. A more refined metric will be learned during the training phase of the GP emulator. This hypothesis is anchored in the interpretation of the climatic precession, where a large positive value signifies an excess of insolation in the Southern Hemisphere during the austral summer, and a large negative value indicates the same in the Northern Hemisphere during the boreal

summer. Moreover, our prior judgement is that the globally averaged temperature responds by the same variation to every doubling of carbon dioxide concentration, hence the logarithm. Accordingly, we also define the rescaled $CO_2$ parameter:

$$m = \log_2\left(\frac{pCO_2}{280\,\text{ppm}}\right). \tag{6}$$

## 2.3 Construction and optimisation of the training set

To construct the training set for the GP emulator, it is essential to sample the parameter space effectively to cover all regions of interest. For this reason, we use Latin Hypercube Sampling (LHS, Eglajs and Audze, 1977; Iman et al., 1981; McKay et al., 2000). This method divides the range of each input parameter into equally probable intervals and selects one sample from each interval. Unlike direct random sampling, LHS provides a more uniform and comprehensive exploration of the parameter space, reducing redundancy and improving the representativeness of sampled points. This method is widely recognized for enhancing emulator accuracy and robustness (Urban and Fricker, 2010).

In this study, the LHS was further optimised using the Audze-Eglājs (AE) criterion (Eglajs and Audze, 1977; Eliáš and Vořechovský, 2016). The AE criterion was developed to optimise the spatial arrangement of points in a hypercube. This is achieved by expressing the potential energy of a system of particles where each pair of particles exerts a repulsive force on each other. These forces are functions of the distance between the pairs of points. By minimising this potential energy, the AE criterion ensures a uniform and space-filling distribution of experimental points.

The optimisation was executed in Julia, a high-level general-purpose dynamic programming language designed for high-performance numerical analysis and computational science (Bezanson et al., 2012), using a genetic algorithm (Bates et al., 2012; Urquhart et al., 2020).

For our application, we must also account for some specificities of the input parameters. Indeed, uniformly sampling the climatic precession and coprecession would lead to unwanted high eccentricity experiments. This is resolved by making a change of variables and sampling uniformly $\{e^2, \varpi, \varepsilon, m\}$. Doing so also requires adapting the distance function in the optimisation algorithm from Euclidean to

$$d_{ij} = \tilde{e}_i^2 + \tilde{e}_j^2 - 2\tilde{e}_i\tilde{e}_j\cos\left(\varpi_i - \varpi_j\right) + \left(\tilde{\varepsilon}_i - \tilde{\varepsilon}_j\right)^2$$
$$+ \left(\tilde{m}_i - \tilde{m}_j\right)^2 \tag{7}$$

where the tilde signifies that the value has been linearly rescaled to [0, 1] interval for the eccentricity and [−1, 1] for the obliquity and $CO_2$. This is necessary because the relative importances of eccentricity-modulated climate precession and obliquity for our variables of interest are a priori unknown. Ultimately, our input space will be optimally filled;

climatic precession and coprecession will follow a semicircular distribution, while obliquity and $m$ will be uniformly distributed.

In this work, based on the computational cost of HadSM3 and previous similar studies (Bounceur et al., 2015; Lord et al., 2017; Van Breedam et al., 2021), the number of samples has been arbitrarily fixed to 81.

## 3 Emulator development

The function of the emulator is to predict the output of HadSM3 for any input in the range spanned by the set produced with the procedure described in Sect. 2.3. Fundamentally, it is an interpolator: if it is well-designed and successfully validated, there is no need for further experiments using the GCM.

### 3.1 Theoretical basis of a Gaussian Process regression

We follow Gaussian Processes (GPs) (MacKay, 1998; Rasmussen, 2006), and specifically, Gaussian Process Regression (GPR) to execute the interpolation procedure.

A GP (MacKay, 1998; Rasmussen, 2006) is a stochastic process, generally comprising random variables indexed by time or space. Its unique property is that any finite collection of these variables adheres to a multivariate Gaussian distribution. Therefore, a GP is a distribution over functions with a continuous domain, effectively describing a probability distribution over an infinite-dimensional vector space.

Let $\mathcal{Z} \subseteq \mathbb{R}^{n_z}$ be the index set, which is the number of input variables (in our case, $n_z = 4$). The function $f_{\text{GP}}(z)$ with $z \in \mathcal{Z}$ is a random variable. A GP is a stochastic process completely described by a mean function $m : \mathcal{Z} \to \mathbb{R}$ and a covariance function $k : \mathcal{Z} \times \mathcal{Z} \to \mathbb{R}$ such that

$$f_{\text{GP}}(z) \sim \mathcal{GP}(m(z), k(z, z')) \tag{8}$$
$$m(z) = \mathbb{E}\left[f_{\text{GP}}(z)\right] \tag{9}$$
$$k(z, z') = \mathbb{E}\left[\left(f_{\text{GP}}(z) - m(z)\right)\left(f_{\text{GP}}(z') - m(z)'\right)\right]. \tag{10}$$

The covariance function serves as a measure of the correlation between two states $(z, z')$, and within the framework of GPs, is known as the kernel.

In Bayesian inference, the GP serves as the prior probability distribution, which can be used for function regression. This Bayesian approach merges new data with existing knowledge. Specifically, Bayes' theorem is employed to integrate the prior with new data, resulting in a posterior distribution. The new data is represented as a training dataset $\mathcal{D} = \{\mathbf{X}, \mathbf{Y}\}$ where $\mathbf{X} = [x_{\text{dat}}^{\{1\}}, ..., x_{\text{dat}}^{\{n_{\mathcal{D}}\}}] \in \mathcal{Z}^{1 \times n_{\mathcal{D}}}$ (in our case $n_{\mathcal{D}} = 81$) is the input matrix and $\mathbf{Y} = [\tilde{y}_{\text{dat}}^{\{1\}}, ..., \tilde{y}_{\text{dat}}^{\{n_{\mathcal{D}}\}}]^{\top} \in \mathbb{R}^{n_{\mathcal{D}} \times 1}$ contains the output values. The one-dimensional regression problem takes the following form

$$\tilde{y}_{\text{dat}}^{\{i\}} = f_{\text{GP}}\left(x_{\text{dat}}^{\{i\}}\right) + \epsilon \tag{11}$$

for all $i \in \{1, \ldots, n_{\mathcal{D}}\}$ and $\epsilon \sim \mathcal{N}(0, \nu^2)$ is i.i.d. Gaussian measurement noise.

Considering that each finite subset of a GP follows a multivariate Gaussian distribution, we can construct the joint distribution for a random test point $z^* \in \mathcal{Z}$:

$$
\begin{bmatrix} \mathbf{Y} \\ f_{\mathrm{GP}}(z^*) \end{bmatrix} \sim \mathcal{N}
$$

$$
\left( \begin{bmatrix} m\left(\boldsymbol{x}_{\mathrm{dat}}^{\{1\}}\right) \\ \vdots \\ m\left(\boldsymbol{x}_{\mathrm{dat}}^{\{n_{\mathcal{D}}\}}\right) \\ m(z^*) \end{bmatrix}, \begin{bmatrix} \mathbf{K}(\mathbf{X}, \mathbf{X}) + \nu^2 \mathbf{I}_{n_{\mathcal{D}}} & \mathbf{k}(z^*, \mathbf{X}) \\ \mathbf{k}(z^*, \mathbf{X})^{\top} & k(z^*, z^*) \end{bmatrix} \right).
$$

(12)

In this expression, the matrix function $\mathbf{K} : \mathcal{Z}^{1 \times n_{\mathcal{D}}} \times \mathcal{Z}^{1 \times n_{\mathcal{D}}} \to \mathbb{R}^{n_{\mathcal{D}} \times n_{\mathcal{D}}}$, is called the covariance or Gram matrix and is related to the kernel via $K_{j,l} = k(\mathbf{X}_{:,l}, \mathbf{X}_{:,j})$ for all $j, l \in \{1, \ldots, n_{\mathcal{D}}\}$.

The posterior predictive distribution of $f_{\mathrm{GP}}(z^*)$ is obtained by conditioning on the test point and the dataset $\mathcal{D}$, using Bayes' theorem:

$$
\mathrm{p}\left(f_{\mathrm{GP}}(z^*) | z^*, \mathcal{D}\right) = \frac{\mathrm{p}\left(f_{\mathrm{GP}}(z^*), \mathbf{Y} | \mathbf{X}, z^*\right)}{\mathrm{p}(\mathbf{Y} | \mathbf{X})}.
$$

(13)

The conditional posterior is a Gaussian distribution described by the mean and variance:

$$
\mu\left(f_{\mathrm{GP}}(z^*) | z^*, \mathcal{D}\right) = m(z^*) + \mathbf{k}(z^*, \mathbf{X})^{\top} \left(\mathbf{K}(\mathbf{X}, \mathbf{X}) + \nu^2 \mathbf{I}_{n_{\mathcal{D}}}\right)^{-1} \left(\mathbf{Y} - \left[m(\mathbf{X}_{:,1}), \ldots, m(\mathbf{X}_{:,n_{\mathcal{D}}})\right]^{\top}\right)
$$

(14)

$$
\mathbb{V}\mathrm{ar}\left(f_{\mathrm{GP}}(z^*) | z^*, \mathcal{D}\right) = k(z^*, z^*) - \mathbf{k}(z^*, \mathbf{X})^{\top} \left(\mathbf{K}(\mathbf{X}, \mathbf{X}) + \nu^2 \mathbf{I}_{n_{\mathcal{D}}}\right)^{-1} \mathbf{k}(z^*, \mathbf{X}).
$$

(15)

A comprehensive detail of the derivation of these formulae can be found in Appendix A of Beckers (2021).

The prior mean function $m(\cdot)$ characterises the average function under the GP distribution before any data is observed. This function presents a simple approach to include prior knowledge about the function we desire to model. An idealized physics model can hence be used as a prior. The GP then models the difference between this prior and the empirical data. When there is no such prior knowledge, a typical choice is to set the prior mean function to zero, i.e., $m(\cdot) \equiv 0$, but following our considerations about the physical climate response discussed in Sect. 2.3, we rather adopt an affine mean function of the form

$$
m(\boldsymbol{x}) = \beta_0 + \boldsymbol{\beta} \cdot \boldsymbol{x} + \beta_e \sqrt{x_1^2 + x_2^2}
$$

(16)

where the last term accounts for a possible effect of the eccentricity. The coefficients $\beta_i$ are chosen to minimise the sum of the squares of the residuals.

## 3.2 Application to climate fields

The regression approach discussed in the previous section is only applicable for sampling one-dimensional functions. However, when we model the GCM outcomes similarly to $\mathbf{f}_{\mathrm{GCM}}(z) + \boldsymbol{\Sigma}$, we are confronted with a dimensionality challenge as the function's image is now of high dimension (a dimension equal to the number of grid points, or pixels). There are several strategies to manage this: using a Multi-Output Gaussian Process (MOGP) or an *outer-product* emulator (Rougier et al., 2009), creating a different GP for each dimension, and/or applying a dimensionality reduction technique (Holden and Edwards, 2010; Wilkinson, 2010). Although a MOGP could theoretically yield better results, its implementation for a GCM longitude-latitude grid is computationally challenging. The design of an appropriate kernel function would be highly complex, and the computational cost of GPR grows cubically with the number of samples ($\mathcal{O}(n_{\mathcal{D}}^3)$), making this approach impractical. In our study, we chose the third strategy and used Principal Component Analysis (PCA) as the dimensionality reduction technique. This choice was driven by our goal of preserving as many teleconnections within the data as possible. Constructing a different GP for each pixel would eliminate many of these correlations. Moreover, the statistical interpretation of the principal components is advantageous, as they can be interpreted as either signal or noise. Furthermore, as the principal components are designed to be uncorrelated with each other, the use of a MOGP appears superfluous.

## 3.3 Kernel function

Equation (14) shows that the selection of the kernel function may significantly influence the posterior mean. The kernel is crucial in expressing how the output associated with two neighbouring inputs are related. Hence, an inappropriate kernel choice or design can result in less accurate, or even incorrect, output (Rasmussen, 2006).

Selecting the appropriate function design with the correct number of hyperparameters is a complex task, but some studies offer an automated method for this search (Duvenaud et al., 2013). The most popular covariance function is the squared exponential kernel (SE-Kernel), as favoured by Kennedy and O'Hagan (2000) and Higdon et al. (2008).

In this study, we initially faced some issues when applying existing models to the Devonian dataset and decided to search for a more general kernel function compared to Bounceur et al. (2015), Lord et al. (2017) and Van Breedam et al. (2021). We evaluated two kernel approaches and found performance differences that are quantified in Sect. 3.6. We settled on the work of Duvenaud et al. (2011) and employed an Additive Kernel, that presented slightly better performance. The kernel works as follows:

To each dimension $i \in \{1, \ldots, n_z\}$, we can assign a *base kernel* [TS3] $k_i(\cdot, \cdot)$. An Additive Kernel is then the sum

$$k_{\text{add}}(z, z') = \sum_{n=1}^{n_z} k_n(z, z') \tag{17}$$

where the first, second and $n$th terms are given by

$$k_1(z, z') = \sigma_1^2 \sum_{i=1}^{n_z} k_i\left(z_i, z_i'\right) \tag{18}$$

$$k_2(z, z') = \sigma_2^2 \sum_{i_1=1}^{n_z} \sum_{i_2=i_1+1}^{n_z} k_{i_1}\left(z_{i_1}, z_{i_1}'\right) k_{i_2}\left(z_{i_2}, z_{i_2}'\right) \tag{19}$$

$$k_n(z, z') = \sigma_n^2 \sum_{1 \leq i_1 < i_2 < \cdots < i_n \leq n_z} \prod_{d=1}^{n} \tilde{k}_{i_d}\left(z_{i_d}, z_{i_d}'\right) \tag{20}$$

where $n_z$ is the dimension of the input space and $\sigma_n \in \mathbb{R}_0^+$. Specifically, the $n$th order interaction kernel $k_n$ is the sum of $\binom{n_z}{n}$ terms. The relative magnitude of each $\sigma_n$ informs about the importance of each order of interaction.

In this research, we have selected the Square Exponential Kernel (SE-Kernel) as the one-dimensional basis function:

$$\tilde{k}_{\text{SE}}(x, y; l) = \exp\left(-\frac{(x - y)^2}{2l^2}\right) \tag{21}$$

with $l \in \mathbb{R}_{>0}$ the lengthscale hyperparameter.

This kernel exhibits notable properties: it is stationary, meaning that it depends only on the distance between each point, and it is universal, implying that a GPR with such a kernel can approximate any continuous function with arbitrary precision on a compact set. Moreover, a SE-Kernel of high dimension can be seen as the product of multiple one-dimensional SE-Kernels.

The entire set of hyperparameters for the Additive Kernel is then defined by the $n_z = 4$ order variance $\{\sigma_1, \ldots, \sigma_{n_z}\}$ and lengthscales $\{l_1, \ldots, l_{n_z}\}$. These define a vector $\boldsymbol{\theta}$.

The first and last order terms in the sum are given by:

$$k_1(z, z') = \sigma_1^2 \sum_{i=1}^{n_z} \exp\left(-\frac{\left(z_i - z_i'\right)^2}{2l_i^2}\right) \tag{22}$$

$$k_{n_z}(z, z') = \sigma_{n_z}^2 \exp\left(-\frac{1}{2} \sum_{i=1}^{n_z} \frac{\left(z_i - z_i'\right)^2}{l_i^2}\right) \tag{23}$$

This Additive Kernel can be seen as an augmented SE-Kernel, as used in Lord et al. (2017). It maintains its properties and computational advantages while adding an optimal level of complexity. This complexity is enough to grasp useful information at a greater distance (see Fig. 5), but still small enough to avoid over-fitting, classically known to appear when parameters to fit are too numerous.

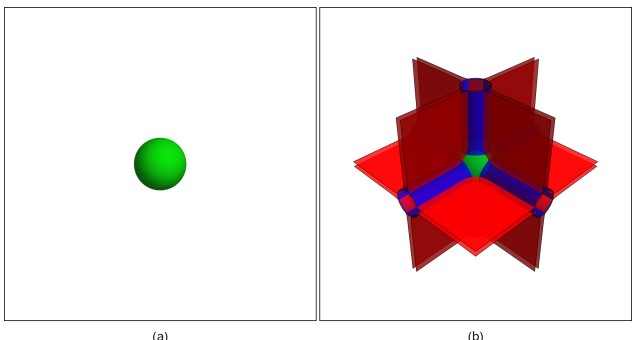

**Figure 5.** Isocontours of **(a)** a Square Exponential Kernel and **(b)** an Additive Kernel in a 3-dimensional space. The SE-Kernel is designed to give importance only to points in close proximity. In contrast, the lower-order kernels (in red and blue) are more inclusive. Reproduced from Duvenaud et al. (2011).

### 3.4 Objective function and optimisation

The GPR being non-parametric but the kernel function possesses free parameters, known as hyperparameters, that must be optimised to best fit the data.

The negative log marginal likelihood function (Eq. 24), as fully derived in Rasmussen (2006), serves as an objective measure of model fit. It quantifies the likelihood of observing the given data under specific hyperparameters. The goal is to minimise this function to find the hyperparameters that maximise the model's likelihood of generating the observed data.

$$\log p(\mathbf{Y}|\mathbf{X}, \boldsymbol{\theta}) = -\frac{1}{2} \tilde{\mathbf{Y}}^\top \left(\mathbf{K}(\mathbf{X}, \mathbf{X}; \boldsymbol{\theta}) + \nu^2 \mathbf{I}_{n_{\mathcal{D}}}\right)^{-1}$$
$$\tilde{\mathbf{Y}} - \frac{1}{2} \log |\mathbf{K}(\mathbf{X}, \mathbf{X}; \boldsymbol{\theta}) + \nu^2 \mathbf{I}_{n_{\mathcal{D}}}| - \frac{1}{2} n_{\mathcal{D}} \log 2\pi \tag{24}$$

where we defined $\tilde{\mathbf{Y}} = \mathbf{Y} - [m(X_{:, 1}), \ldots, m(X_{:, n_{\mathcal{D}}})]^\top$ to take into account a non-zero mean-function.

$$\begin{bmatrix} \boldsymbol{\theta}_{\text{opt}} \\ \nu_{\text{opt}} \end{bmatrix} = \operatorname{argmin}_{\boldsymbol{\theta}, \nu} - \log p(\mathbf{Y}|\mathbf{X}, \boldsymbol{\theta}) \tag{25}$$

The negative log marginal likelihood function effectively balances model complexity and goodness of fit.

Optimising the negative log marginal likelihood provides an approach to hyperparameter tuning with a theoretical justification. Furthermore, by design, it includes a complexity penalty that discourages overly complex models, effectively preventing overfitting and thus promoting the selection of parsimonious models.

Yet, its optimisation can be computationally intensive, and complicated by the presence of multiple local optima.

With these difficulties in mind, we implemented the GPR in the Julia language using the versatile AbstractGP.jl package (Widmann et al., 2024) and Zygote.jl (Innes, 2019) to efficiently compute the gradients. The optimisation scheme

is based on a bounded probabilistic descent and a local L-BFGS gradient descent algorithm, both available within the Optimization.jl package (Dixit and Rackauckas, 2023). This method showed the best performance amongst common algorithms. We initiate the algorithm from a wide range of initial conditions to favour a global minimum. Since the values to emulate are rescaled to be of order 1, we expect the lengthscale and order-variance parameters to be of the same order. We first run 1024 optimisations at low tolerance from points lying in a Latin hypercube of bounds $[10^{-16}, 150]$ for each $l_i$, $[0, 2]$ for each $\sigma_i$ and $[10^{-6}, 2]$ for $\nu^2$, where a nonzero lower bound ensures numerical stability. The 64 lowest different minima are kept and used in a second optimisation round with a much lower tolerance. This procedure strongly favours a global optimum.

## 3.5 Data preprocessing

As discussed in Sect. 3.2, the GPR is not directly applied to the climate fields but rather to the PCA loading coefficients, building an independent GP for each component. PCA condenses the variability of multivariate observations into a set of orthogonal modes, known as principal components (PCs). These PCs encapsulate the most significant patterns in the data. The mathematical foundation of PCA involves the computation of eigenvalues and eigenvectors from the data's covariance matrix. The eigenvectors, also termed as loadings, signify the directions of maximum variance, while the eigenvalues quantify the variance explained along each eigenvector. These loadings offer insights into the relationships amongst the original variables and assist in identifying the primary modes of variability within the dataset.

The decomposition of the output in PCs is a linear operation (although the PCs themselves are non-linear), as illustrated by the following equations:

$$y = \mathbf{P}_{\mathrm{GCM}}^{\top} (x - \mu_{\mathrm{GCM}}) \tag{26}$$
$$\tilde{x} = \mathbf{P}_{\mathrm{GCM}} y + \mu_{\mathrm{GCM}}. \tag{27}$$

In Eq. (26), $\mathbf{P}_{\mathrm{GCM}}$ denotes the matrix of loadings, whose columns are the orthonormal eigenvectors of the data. The term $x$ signifies a climate field that has been reshaped into a vector and imputed for missing values, while $\mu_{\mathrm{GCM}}$ represents the mean value of each gr $x$ onto the lower-dimensional vector space spanned by the PCs, the components of $y$ are often referred to as the PCA scores. These scores are the scalar values interpolated using the GPR.

Equation (27) demonstrates how to reconstruct the climate field from the lower-dimensional representation (denoted by $\tilde{x}$). This reconstruction is approximate and will only be exact if the true value $x$ resides within the vector space spanned by the data used to generate the projection matrix. This equation is employed to reconstruct the GCM outcomes using the results from the GPR.

For optimal performance of PCA, certain data assumptions must be satisfied. Specifically, the data should follow a Gaussian distribution to ensure the validity of statistical measures like covariance. Standardizing (i.e., subtracting the mean and dividing by the standard deviation) is also a common procedure for ensuring comparability in scale across pixels, preventing dominance by points with larger magnitudes.

Standardisation of each pixel is typically applied to data that follows a Gaussian distribution, which is the case for our temperature data. Precipitation minus evaporation (or *runoff*, which often contains many zeros in arid grid cells) is conversely highly asymmetric and skewed. In that case, standardisation does not yield a Gaussian distribution via affine transformation of the data.

In this study, we chose to standardise temperature but not runoff before computing the PCA. This proved to be the most effective approach. Additionally, we opted to emulate the total annual water volume (i.e., runoff multiplied by area) rather than runoff directly. This decision is motivated by the unequal areas of grid cells. As a result, this procedure effectively assigns greater weight to tropical regions, which are of particular interest in our study.

The PCA scores themselves are also rescaled. Indeed, the magnitude of these scores is proportional to $\sqrt{\lambda}$, where $\lambda$ is the variance associated with the principal component of interest (i.e., an eigenvalue). To ensure the stability of the optimisation algorithm and the interpretability of the objective function, we divide each coefficient by $2\sqrt{\lambda}$. This transformation ensures that the values to be interpolated typically lie between $-1$ and $1$ and have the same scale across different principal components.

In principle, some experiments may exhibit a highly nonlinear behaviour, e.g. by crossing some tipping points, and effectively behave as outliers. When such outliers occur, they deserve particular attention because they may complicate the learning process and justify a specific treatment.

The adopted strategy to detect these points is based on the PCA decomposition of the set. We assume that each component of $y$ follows a Gaussian distribution. The mean and standard deviation of each principal component are computed while excluding a specific GCM outcome. This outcome is then projected onto the lower-dimensional vectorial space spanned by the remaining data, and its $z$-score is computed as follows:

$$z_{\mathrm{score},k} = \left| \frac{\tilde{y}_k^{\{i\}} - \mu_k^{\{-i\}}}{\sigma_k^{\{-i\}}} \right|. \tag{28}$$

Here, the superscript $\{i\}$ is the experiment index, $\{-i\}$ signifies that $\mu$ and $\sigma$ have been computed excluding the specific experiment, and the subscript $k$ is the PC dimension. If the value of this score is higher than a specific threshold (e.g., 3 for a 99.73 % accuracy) for the main contributing PCs, then the probability that the behaviour of this GCM run is an outlier is high and should be excluded, or at least specifically examined. In our set of 81 HadSM3 runs, though, none of

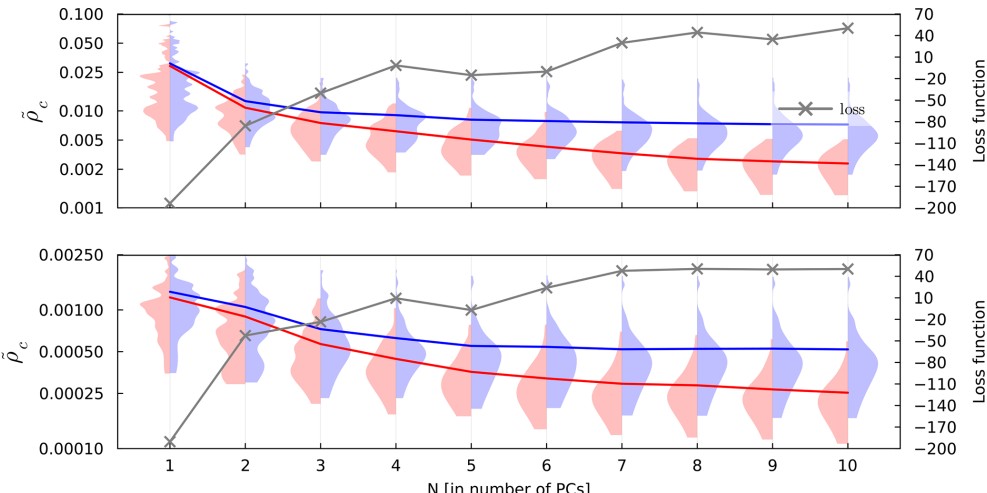

**Figure 6.** Shifted concordance correlation coefficient ($\tilde{\rho}_c = 1 - \rho_c$) distributions (left axis, in log-scale) and individual loss function value (right axis) for runoff (top) and surface temperature (bottom). In blue, the CCC has been computed on a sample while having excluded it during the learning process (*leave-one-out*; LOO). In red are depicted the CCC using the full set of experiments. Plain blue and red lines represent the average value across the whole set of 81 GCM runs, while the grey line indicates the objective function value for the specific $N$th PC.

them had to be excluded and no specific treatment was thus necessary.

### 3.6 Validation

The last step in our process is to determine the best number of PCs to emulate. Insufficient PCs could result in information loss, while an excess might introduce noise, given that the least significant components often resemble random fields attributable to the GCM internal variability.

In this study, we opted for a dimensionless metric: the weighted Concordance Correlation Coefficient (CCC) (Lin, 1989), to quantify the accuracy of emulator-based reconstructions.

$$\rho_c(\boldsymbol{y}, \hat{\boldsymbol{y}}) =$$
$$\frac{2\sum_i w_i \left(y_i - \mu_y\right)\left(\hat{y}_i - \hat{\mu}_y\right)}{\sum_i w_i \left(y_i - \mu_y\right)^2 + \sum_i w_i \left(\hat{y}_i - \hat{\mu}_y\right)^2 + \left(\mu_y - \hat{\mu}_y\right)^2} \quad (29)$$

where $\hat{y}_i$ denotes the prediction, and $y_i$ the GCM output. This approach involves excluding a single data point during training and using it to validate the emulator's prediction. The calculated CCC values, combined with the loss function, guide the selection of the optimal number of PCs to retain for each variable.

As indicated by Fig. 6, retaining the first 7 PCs for both runoff and surface temperature achieves an effective balance between minimising information loss and avoiding noise amplification. This compromise ensures the emulator's robustness and reliability for climate field predictions.

We can also use the same metric to evaluate the effectiveness of a more complex kernel function. To do so, we compare two emulators designed to be as similar as possible; their only difference lies in the correlation function $k(z, z')$. For one emulator, this function is an *Additive Kernel* as described earlier, while for the other it is the more conventional *Square Exponential kernel*, as used in previous studies (Bounceur et al., 2015; Lord et al., 2017; Van Breedam et al., 2021). Figure 7 presents the results of this comparison for continental runoff – the variable that initially motivated the development of this improved emulator because of its initial poor predictability. We observe that employing the Additive Kernel function results in superior performance for almost all the test experiments.

### 3.7 Interpretation and sensitivity

Figure 8 shows the first six PCs and the associated normalised variance $\lambda_i/\lambda_{\text{tot}}$, which indicates the proportion of variability each PC explains. As these are variations relative to the mean value and can be multiplied by either a positive or negative factor, the sign is not inherently meaningful. Consequently, the terms "cooling" and "warming" can be interchanged without loss of generality in the subsequent discussion.

The first PC, accounting for approximately 97 % of the total variance, represents a global anomaly, where the sign is consistent across locations, but the intensity varies. As inferred from Fig. 9a, this component primarily results from changes in $p\text{CO}_2$. However, as depicted in Fig. 10a, eccentricity has a similar signature, as expected, since this parameter modulates the total amount of radiation received from the Sun. The second PC depicts a polar effect, with cooling at the

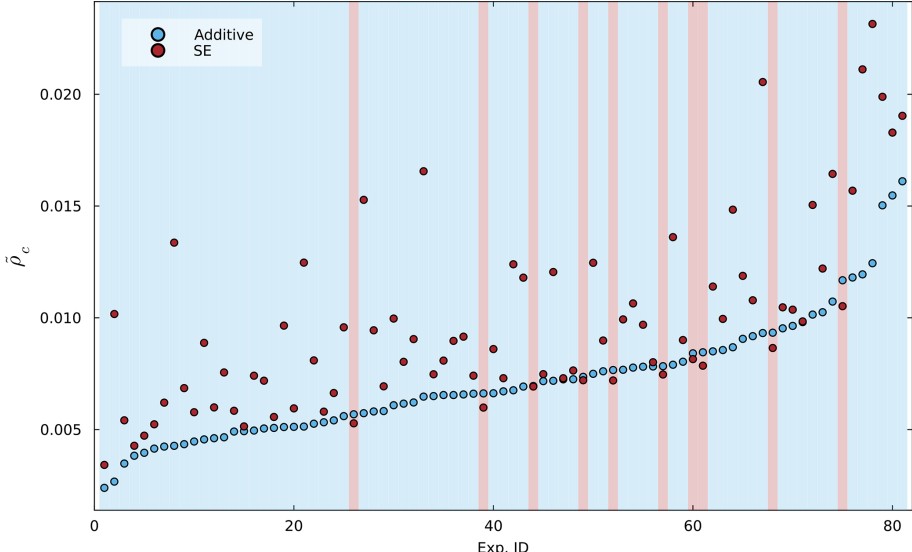

**Figure 7.** Comparison of the performance of the two kernel functions: Additive and Square Exponential (SE). Each point represents the shifted CCC value computed between the GCM climate field and the prediction produced by an emulator trained on the dataset with the specific experiment omitted (LOO). A blue background indicates that the Additive Kernel outperforms the alternative in that experiment, whereas a red background signifies that the SE kernel achieves better performance. The 81 experiments were sorted by $\tilde{\rho}_{c,\text{Additive}}$ for clarity.

poles and warming in the mid-latitudes. As shown in Fig. 9b, this effect is mostly attributable to changes in obliquity.

The subsequent PCs reveal more localised variations. Notably, the sixth PC is highly localised and shows most of its effect in the polar south region. This specific temperature anomaly can be attributed to the presence of sea ice. To capture this particular pattern, it may be beneficial to design a focused emulation process for this region. This approach would increase the variance of this specific component, enabling better handling by the GPR. Visualisation of the remaining PCs shows that this particular area is the only one with a significant anomaly, suggesting that the other PCs are less relevant for the emulation process. A specific design where different GPRs are trained for continental and sea-surface temperature has also been considered, but the performance improvement – it showed slightly better results amongst 55 % of the training set – was not deemed significant enough to justify the additional computational cost when merged inside GEOCLIM. The corresponding principal components for runoff are shown in Supplement Fig. S2.

Maps comparing high vs. low climate precession and high vs. low obliquity for temperature and runoff are provided in Figs. S3–S8.

As described in the previous text, most of the variance arises from changes in $p\text{CO}_2$. Indeed, the experimental design allows for significantly more variance in this direction than in climate precession. However, one could restrict the parameter space accordingly and compute, for instance, the global mean surface temperature (GMST) to test whether the results are consistent with the astronomical forcing or not.

If we assume that a doubling of $\text{CO}_2$ decreases outgoing radiation by $3.7\,\text{W}\,\text{m}^{-2}$, and that the transition from zero eccentricity to $e_{\text{max}}$ is equivalent (in terms of total mean annual radiative forcing) to (Milankovitch, 1941; Berger and Loutre, 1994):

$$\left(\left(1 - e_{\text{max}}^2\right)^{-\frac{1}{2}} - 1\right)\frac{S}{4} \approx 0.93\,\text{W}\,\text{m}^{-2}$$

for $e_{\text{max}} = 0.075$ and $S = 1322\,\text{W}\,\text{m}^{-2}$, then we can estimate that a multiplicative factor of $2^{\frac{1.06}{3.7}} \approx 1.2$ for $p\text{CO}_2$ would produce similar effects on GMST for both astronomical forcing and carbon dioxide.

As shown in Fig. 11, a 20 % variation in $p\text{CO}_2$ results in a GMST change equivalent to the effect of increasing eccentricity from $e = 0$ to $e = 0.075$, corresponding to approximately $+0.7\,°\text{C}$. This suggests that orbital variability is not overshadowed by variations in atmospheric carbon dioxide.

## 3.8 Coupling to GEOCLIM

The GP calibration is done in a high-level computing language, namely Julia, for easier tuning and understanding of the code. However, most of the historical models in climate science are written in Fortran, and this is the case for GEOCLIM. Once trained, making a prediction with the emulator boils down to multiplying matrices, operations that are well optimised in Fortran using the OpenBLAS library. A script to automatically generate the Fortran routine, given a trained emulator, has been written for this coupling purpose.

In our setup, we want the $p\text{CO}_2$ to be interactive and, therefore, cannot precompute the whole climate time series,

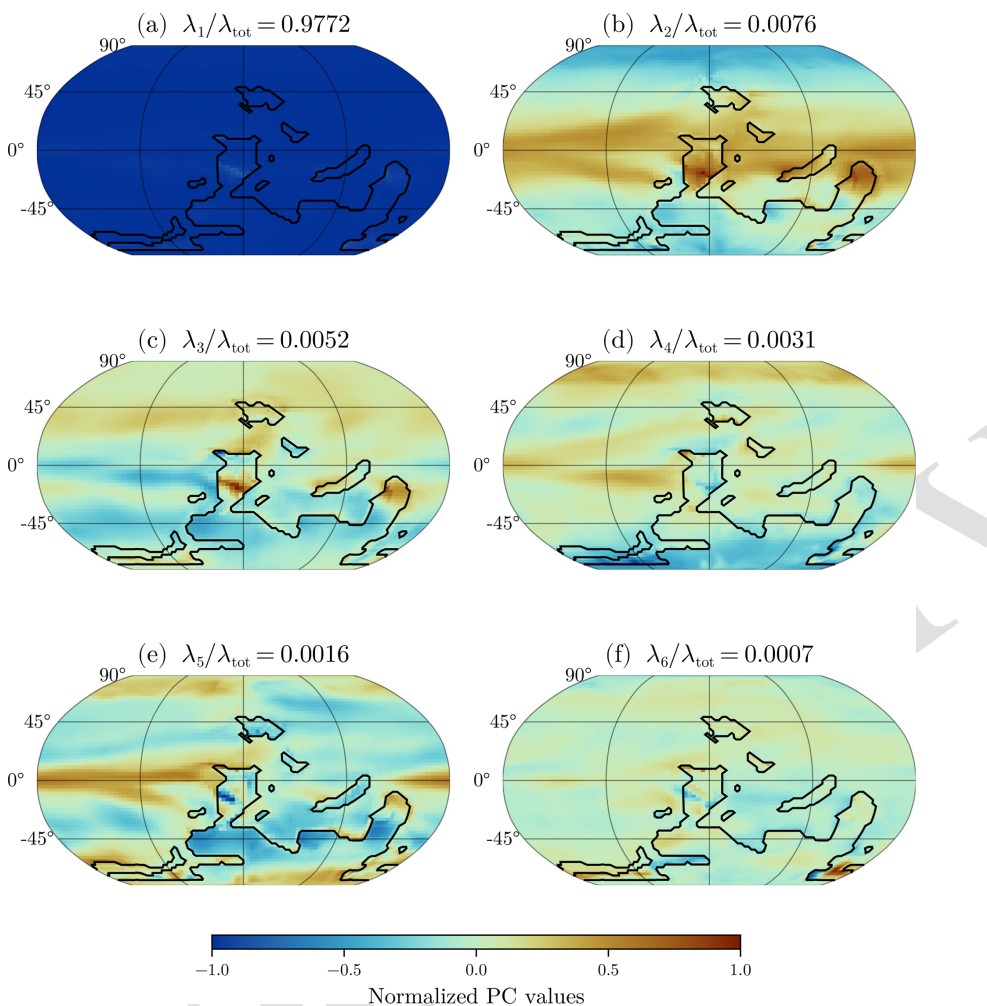

**Figure 8.** The figure presents the first six principal components of the surface temperature, ordered by their eigenvalues and rescaled by a multiplicative factor. The colour scale is indicative of the amplitude and its sign within a single PC, but is independent across different PCs. The black lines delineate the coastlines of the continental configuration.

as we need input from the biogeochemical model to predict the next climate state. The coupling time is an arbitrary parameter that depends on the time resolution desired and the magnitude of climatic perturbations. Depending on the situation, this parameter can be easily adjusted to a different value. Given the processes featured by GEOCLIM and the timescale of astronomical forcing, it is conservative to update the climate fields every 25 years and synchronise it with the calls of the continental weathering module. We do not expect any process to operate on a faster timescale.

We emphasise that our emulator predicts annual-mean temperature and runoff fields, which GEOCLIM uses as inputs. Indeed, GEOCLIM is not designed to simulate seasonal variability. Yet, the seasonal cycle is captured in the HadSM3 simulations used to train the emulator. In particular, orbital forcing effects on the seasonal cycle effectively influence the annual-mean fields that the emulator produces – for example, this is how monsoons respond to precession. In princi-

ple, the emulator could be adapted to account for seasonal effects by emulating monthly mean climate fields, but this would require a significant decrease in the model timestep and a corresponding large increase in computational cost.

The complete emulator development workflow described in this section is summarized in Fig. 12, which illustrates the seven key methodological steps from parameter space design through final integration into GEOCLIM.

## 4 Case study: orbital forcing of the Late Devonian climate

### 4.1 Devonian Ocean Anoxic Events and astronomical forcing

The dynamics of Ocean Anoxic Events (OAEs) and their responsible mechanisms have been investigated over the last decades, but failed to yield consensus (see for instance Had-

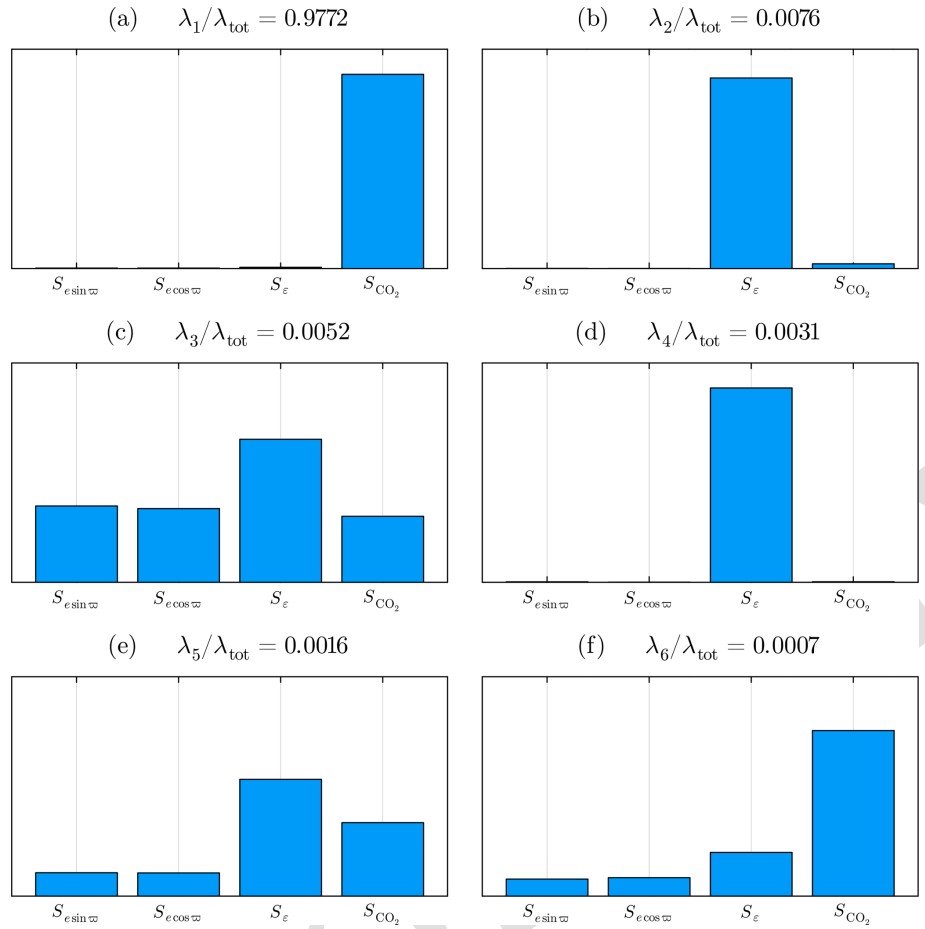

**Figure 9.** Magnitudes of the total Sobol index (Sobol', 2001) for each input variable. A superior total-order Sobol index signifies the parameter's comprehensive influence on the output's variability, integrating both direct impacts and interaction effects. Due to a larger domain explored for the carbon dioxide and obliquity, the effect of climatic precession is screened (cf. Fig. 10).

dad et al., 2018). It has been hypothesized that the cyclical recurrence of anoxia in geological records can be influenced by astronomical cycles within a broader context of background environmental conditions (cf., De Vleeschouwer et al., 2017; Whalen et al., 2017; Da Silva et al., 2020; Lu et al., 2021; Ma et al., 2022, 2025; Wichern et al., 2024). Specific configurations of the astronomical forcing would help triggering regional or even possibly global anoxia in a context of already low-oxygen content. Furthermore, orbital forcing would pace anoxia bursts. One suggested mechanism involves the influence of astronomical forcing on the position of tropical precipitation. Such changes would affect the dynamics of nutrient storage on the continent, as well as the rate and time at which they are released in the oceans. Finally, the modulation of nutrient release would impact ocean biogeochemistry (De Vleeschouwer et al., 2017, 2024; Wichern et al., 2024).

In the specific case of the Devonian, De Vleeschouwer et al. (2017) and Wichern et al. (2024) suggest that conditions for triggering and sustaining anoxia may best be met after passage through a so-called node of the very-long eccentric-ity cycle: a configuration met approximately every 2.4 million years (Laskar, 2020). However, as the precise timing of astronomical forcing during the Paleozoic is uncertain due to the chaotic nature of planetary dynamics, the investigation strategy needs to consider different scenarios. Furthermore, the anoxia itself may have a duration ranging from several tens of thousands to approximately one hundred thousand years, which poses additional questions about sustaining mechanisms. A possible explanation would be a positive feedback loop involving nutrient recycling (Reershemius and Planavsky, 2021; De Vleeschouwer et al., 2024; Wichern et al., 2024).

## 4.2 Calibration of GEOCLIM

GEOCLIM version 6.1.0 was selected for long-term carbon cycle modelling. While the primary modification is the implementation of the emulator for dynamic climate computation, several other adjustments were necessary for our model setup.

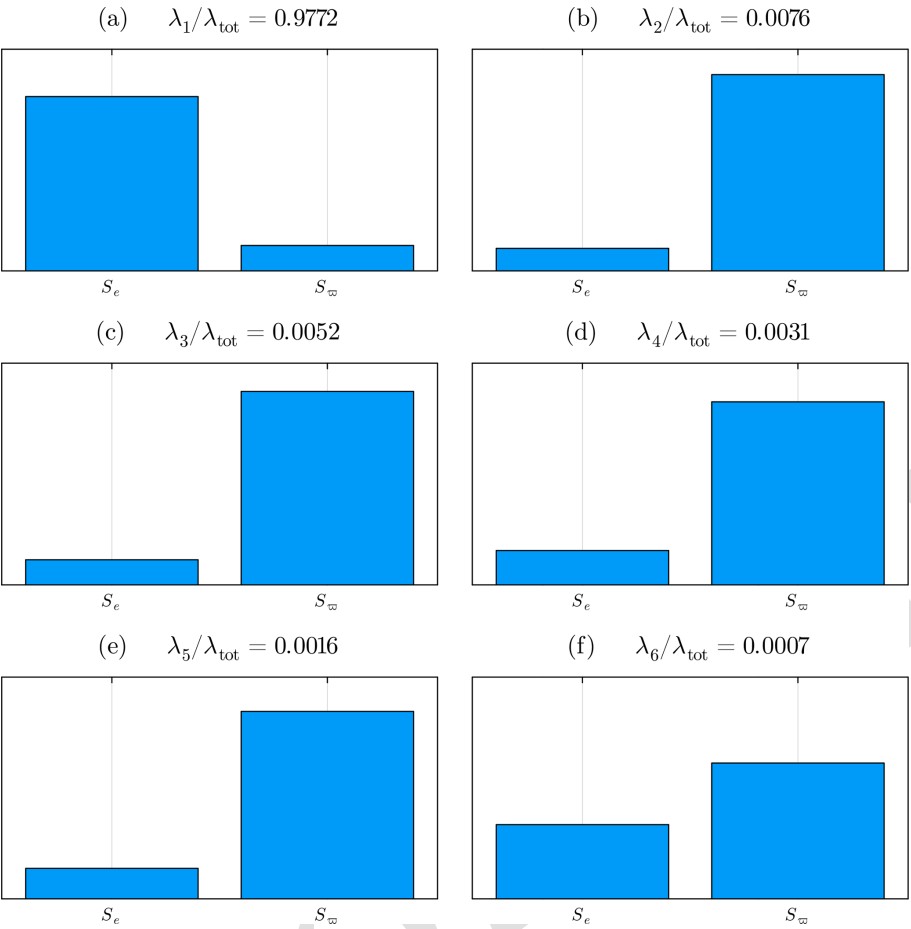

**Figure 10.** Magnitudes of the total Sobol index for eccentricity $e$ and longitude of the perihelion $\varpi$. To obtain this figure, the values of obliquity and $p\text{CO}_2$ were held constant respectively at $\varepsilon = 22.23°$ and $5 \times 280\,\text{ppm}$.

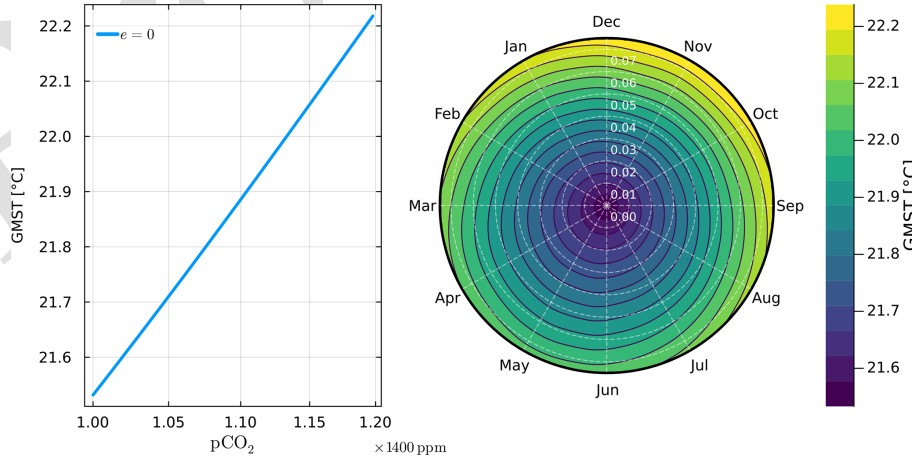

**Figure 11.** The left panel shows the global mean surface temperature (GMST) for a fixed astronomical configuration ($e = 0$, $\varepsilon = 22.23°$) with $p\text{CO}_2$ varying from $5 \times 280\,\text{ppm}$ to $1.2 \times 5 \times 280\,\text{ppm}$. On the right is displayed a phase contour plot of the GMST, with the eccentricity as the radius and the longitude of perihelion as the azimuthal angle. The angle is expressed in the month when the Earth is closest to the Sun, such that the vertical axis is the climatic precession $e \sin \varpi$ and the horizontal axis is the coprecession $e \cos \varpi$. The polar representation is shown for a fixed obliquity ($\varepsilon = 22.23°$) and a constant $p\text{CO}_2$ of $5 \times 280\,\text{ppm}$.

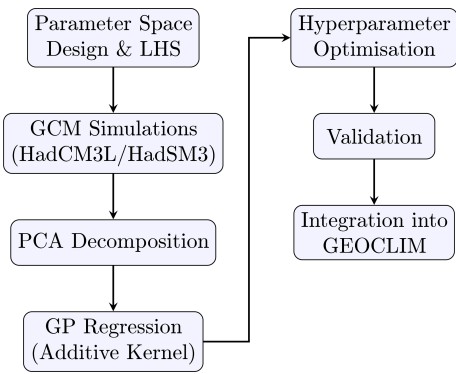

**Figure 12.** Methodological flowchart showing the seven key steps in the emulator development process: (1) parameter space design using Latin Hypercube Sampling, (2) GCM simulations with the two-tier HadCM3L/HadSM3 approach, (3) PCA decomposition of climate fields, (4) GP regression using additive kernel for each PC, (5) hyperparameter optimisation, (6) validation, and (7) integration into GEOCLIM.

The temperature of each oceanic box is determined through a linear relationship with the global sea-surface temperature (GSST):

$$T_i = \gamma_{0,i} + \gamma_{1,i} \times \text{GSST} \tag{30}$$

where the GSST is derived from the emulator output. Since our HadSM3 setup employs a slab ocean model without vertical resolution, we require a method to specify oceanic box temperatures for GEOCLIM. The coefficient vectors $\boldsymbol{\gamma}$ were fitted using least-squares regression on cGENIE simulations with a similar continental configuration (Gérard et al., 2025), following the same experimental design as our 81 HadSM3 runs. This parameterisation ensures that oceanic box temperatures incorporate both orbital forcing and $pCO_2$ fluctuations through the emulated GSST, with updates synchronised to the 25-year climate field intervals.

Water mass exchanges between neighbouring boxes were also updated accordingly, though kept constant in time and across experiments as this version of GEOCLIM is not designed to include a dynamic ocean circulation.

Following the methodology outlined in the supplementary material of Maffre et al. (2021), we calibrated the uncertain parameters using a present-day configuration. Box geometries (volumes and surfaces) were computed using Scotese's reconstruction. The surfaces and volumes of epicontinental boxes, which carry significant uncertainty, were adjusted to achieve an equilibrium concentration of $10 \, \text{mol} \, \text{m}^{-3}$ dissolved $SO_4$ in the ocean (Cai et al., 2022), using mean-orbital parameters ($e = 0$ and $\varepsilon = 22.23°$).

Given the substantial uncertainties in reconstructing solid Earth degassing rates during the Devonian period (Marcilly et al., 2021; Müller et al., 2024), we adopted a fixed value in this study. This value was determined by calculating the degassing rate necessary to balance the total silicate weathering consumption in a present-day configuration, corresponding to the rate required to maintain a steady atmospheric $pCO_2$ level of 280 ppm.

## 4.3 Fourier analysis of the response

The development of this emulator and modelling strategy was motivated by investigating the potential connection between astronomical forcing and the timing of Devonian OAEs, as discussed in the introduction. Previous studies by De Vleeschouwer et al. (2024), and Wichern et al. (2024) hypothesized that reduced seasonal contrasts during eccentricity minima within 2.4 Myr nodes facilitate gradual regolith accumulation on continents. This accumulated regolith is subsequently mobilized and eroded during the first pronounced precession maximum following the node. Our modelling framework enables the investigation of potential long-term effects within geological soil systems.

In this part, we present an integrated result of the emulator into GEOCLIM6.1.0. The astronomical scenario, which outlines the values of the three orbital parameters over time, is a plausible but hypothetical scenario, given that no precise solution for the Devonian can be produced. We use the planetary solution La10a provided by Laskar et al. (2011), and selected the period 125–100 Ma, during which 2.4 Myr eccentricity nodes are well-marked. We coupled this solution with the precession model of Sharaf and Boudnikova (1967), following the procedure outlined in Berger and Loutre (1991). This procedure implies, first, a harmonic decomposition of the planetary solution (using R-code based on Šidlichovský and Nesvorný, 1997) and then specification of reference obliquity and general precession rate. The latter are given by Farhat et al. (2022) for the period of 370 Ma. With this construction, the solution has obliquity frequencies of 30 555 and 37 335 years, and climatic precession frequencies of 16 470 and 19 386 years considered to be plausible for the Devonian. The R-code used to generate those time series is available in Crucifix (2025).

Starting from its calibration value, the model was run for 25 Myr. The last 22.5 Myr were selected for detailed analysis because the static equilibrium state, which has a slightly higher $pCO_2$, differs from the dynamic equilibrium state. The initial 2.5 Myr were used to stabilise the model and avoid an unwanted artefact trend.

Figure 13 shows the normalised Fourier power spectral density of the forcing (i.e., the three orbital parameters) and of a spatial average and latitudinal band of the climate fields, specifically continental temperature and runoff. These fields are internally computed by the emulator based on the orbital parameters and carbon dioxide concentration. Globally, the surface temperature shows a strong influence from the obliquity and eccentricity cycles. Conversely, runoff is primarily affected by climatic precession ($e \sin \varpi$). Locally, the long eccentricity cycle ($\pm 405$ kyr) seems to be dominant, but its influence is reduced when considering the global average.

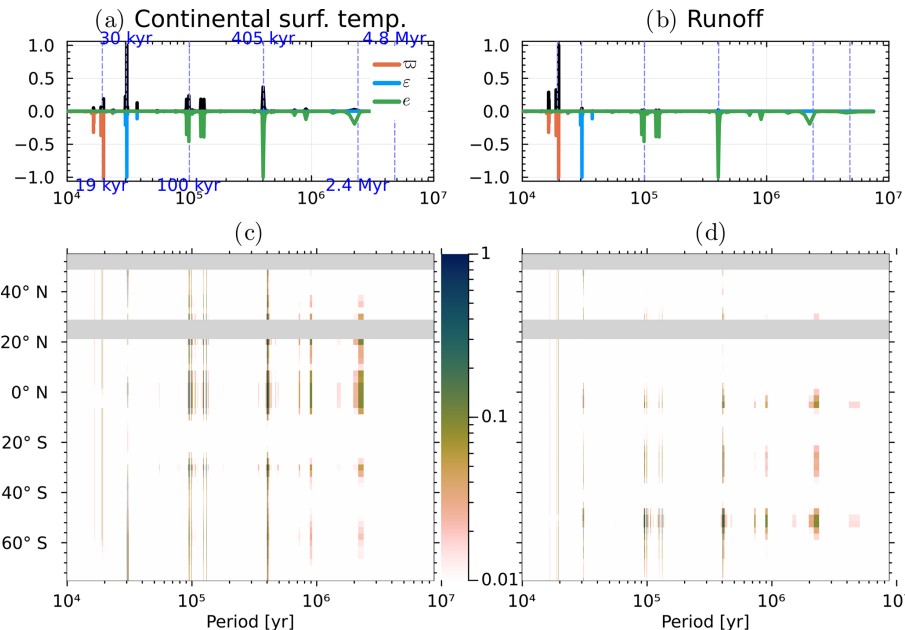

**Figure 13.** Normalised Fourier power spectral density of **(a, c)** continental surface temperature and **(b, d)** runoff. The top panels show the spectrum of the spatial average in black and the spectra of the orbital parameters flipped over the *x* axis (for better visibility) and the bottom panels display the latitudinal averages. Various important cycles are shown in vertical dashed blue lines as landmarks.

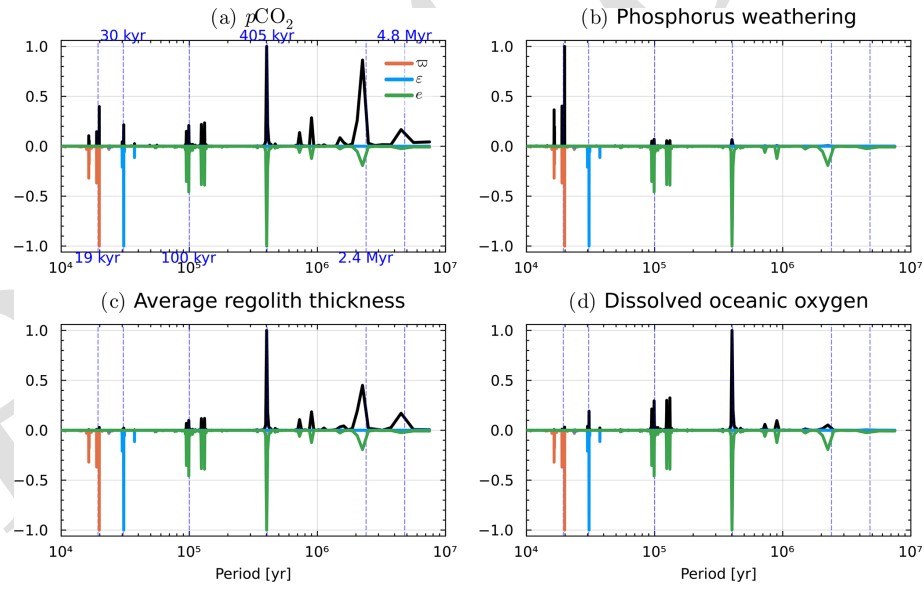

**Figure 14.** Normalised Fourier power spectral density of **(a)** atmospheric $pCO_2$, **(b)** total reactive phosphorus weathering flux, **(c)** global average regolith thickness and **(d)** global oceanic dissolved oxygen. The spectra of the orbital parameters are shown flipped over the *x* axis. Various important cycles are shown in vertical dashed blue lines as landmarks.

The 2.4 Myr cycle is present in the forcing, but its influence on the climatic fields is very weak.

From Fig. 14a and c, we can observe that the atmospheric $pCO_2$ and average regolith thickness exhibits strong 405 kyr and 2.4 Myr cycles, suggesting the presence of nonlinearities in the subsystem. The atmospheric $pCO_2$ varia-

tions shown in the spectral analysis exhibit amplitude variations of up to 50 ppm, which are comparable to the temporal variations displayed in Fig. 17. As depicted in Fig. 15, the amplification of the 2.4 Myr cycle is a spatially localised feature, with a maximal amplitude around the tropics. Time series of GEOCLIM outputs are presented in the following sec-

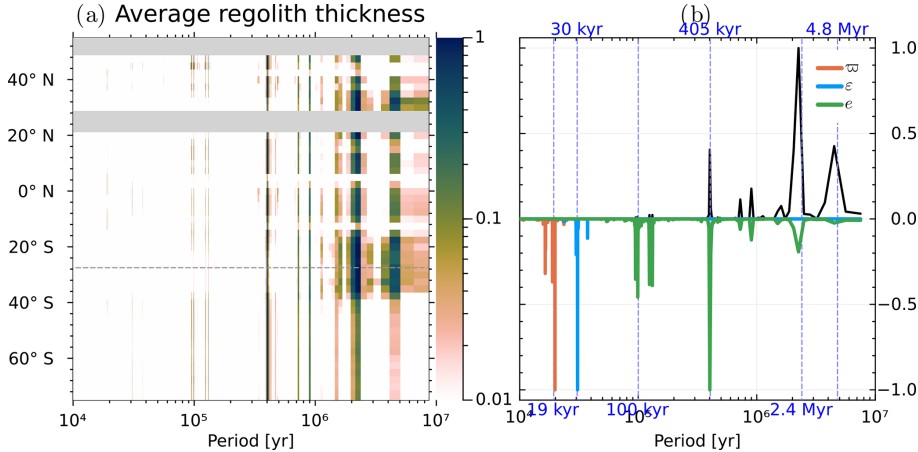

**Figure 15.** Normalised Fourier power spectral density of **(a)** longitudinal average of the regolith thickness and **(b)** a section at 27.5° S, represented by the dashed line in **(a)**. The spectra of the orbital parameters are shown flipped over the $x$ axis. Various important cycles are shown in vertical dashed blue lines as landmarks.

tion to further investigate this local phenomenon. However, these cycles are almost absent from the total reactive phosphorus flux, as shown in Fig. 14b. Under the parametrisation of GEOCLIM, this result is inconsistent with the hypothesis that the regolith thickness is a major driver of the nutrient weathering flux.

An illustrative time series of oceanic dissolved oxygen for one scenario is shown in Fig. S9.

## 4.4 Regolith dynamics

Here, we examine the variations in regolith thickness within the model. Regolith thickness is a crucial parameter in GEO-CLIM, controlling weathering rates and governing the efficiency of continental weathering processes. As such, it is central to many hypotheses regarding Devonian OAEs.

The simulation employs the same astronomical scenario as previously described, but with obliquity held constant at 22.23° to isolate the effects of eccentricity and climatic precession. Figure 17 presents a 2 Myr segment of this scenario, displaying three model outcomes: atmospheric carbon dioxide concentration, average runoff, and surface temperature in two distinct regions illustrated in Fig. 16.

Figure 17 shows a clear precession signal on the runoff and surface temperatures. Nonetheless, the amplitude and duration of these perturbations alone are insufficient to independently trigger a global OAE. As depicted in Fig. 18, the regolith in the green region predominantly maintains equilibrium and does not exhibit significant thickening paced by eccentricity. Conversely, the blue region demonstrates growth paced by this parameter. The green region can be characterized as an ever-wet tropical region, whereas the blue region experiences aridity paced by precession minimum.

In the blue region, when eccentricity is high, the moisture available during a precession maximum amplifies re-

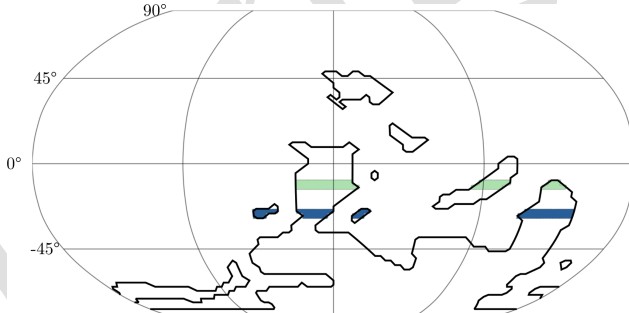

**Figure 16.** The two regions under consideration for the spatial averages in Fig. 17. The green region is consistently wet, while the blue region, located at the edge of the intertropical convergence zone (ITCZ), can experience arid conditions under precession minima.

golith production more than it augments erosion, resulting in a thickening of the regolith. In the ensuing precession minimum, the absence of water for production or erosion renders the regolith static until the next precession maximum for further growth. This mechanism operates as long as the eccentricity increases. However, under conditions of decreasing eccentricity, the subsequent precession maximum lacks sufficient wetness to sustain the thickness of the regolith, leading to production becoming lower than erosion. A remarkable phenomenon can be observed during an eccentricity node. During this period, the eccentricity is low for a complete 405 kyr cycle, and the regolith thickness at the beginning, resulting from previous high eccentricity cycles, is more important than its equilibrium value. This results in a significantly higher erosion than production of the regolith during the node, creating an important 2.4 Myr cycle in the signal thickness.

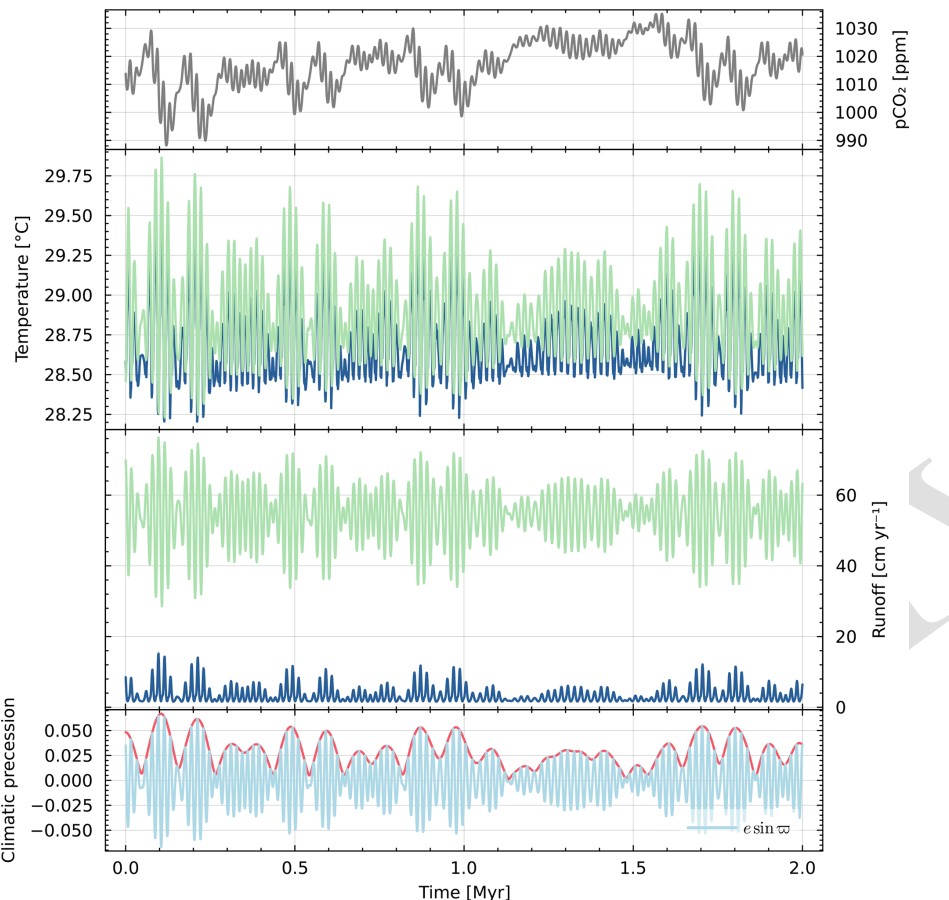

**Figure 17.** This figure displays a 2 Myr long segment of a plausible Devonian orbital solution with constant obliquity. The top line represents the atmospheric $p$CO$_2$, the blue and green curves are spatial averages on the respective regions of the same colour as depicted in Fig. 16. The eccentricity is shown in red on the bottom graph. Except for climatic precession, which is the forcing here, all other curves are outcomes of the model. This section has been chosen to specifically show a 2.4 node, approximately located between 1.1 and 1.5 Myr, where the 100 kyr eccentricity signal is almost absent.

In the green region, both production and erosion are influenced to a comparable degree by precession, resulting in the regolith nearing a steady-state.

From these findings, our model suggests an alternative narrative to the eccentricity node hypothesis. Globally, the regolith (controlling weathering-driven nutrient delivery) demonstrates a positive correlation with eccentricity, indicating that soil development occurs predominantly during high eccentricity periods rather than during nodes. Furthermore, the impact of regolith thickness variations on oceanic nutrient fluxes appears minimal.

As illustrated in Fig. 18, these variations are relatively modest, with runoff – primarily controlled by climatic precession $e \sin \varpi$ – emerging as the dominant driver of weathering and erosion processes. Moreover, the majority of nutrients originate from regions analogous to the green zone, where significant long-term response effects appear absent.

## 4.5 Discussion of the case study

With this case study, we now have a framework for studying the astronomical forcing on regolith dynamics during the Devonian period. With the hypotheses adopted here (fixed palaeography, solid Earth degassing, vegetation and constant lithology), results indicate that regolith build-up is more closely linked to periods of high eccentricity rather than to eccentricity nodes.

More specifically, we found that climatic precession ($e \sin \varpi$) primarily controls runoff and weathering, while the direct impact of regolith dynamics on nutrient fluxes appears minimal. This confirms that, in the Devonian context, additional environmental factors are necessary to trigger widespread OAEs. For example, obliquity may play a role in sea-level variation through effects from ice sheets (Ma et al., 2022, 2025).

In the context of our study, it appears impossible to create an OAE from orbital forcing alone. The link between nutri-

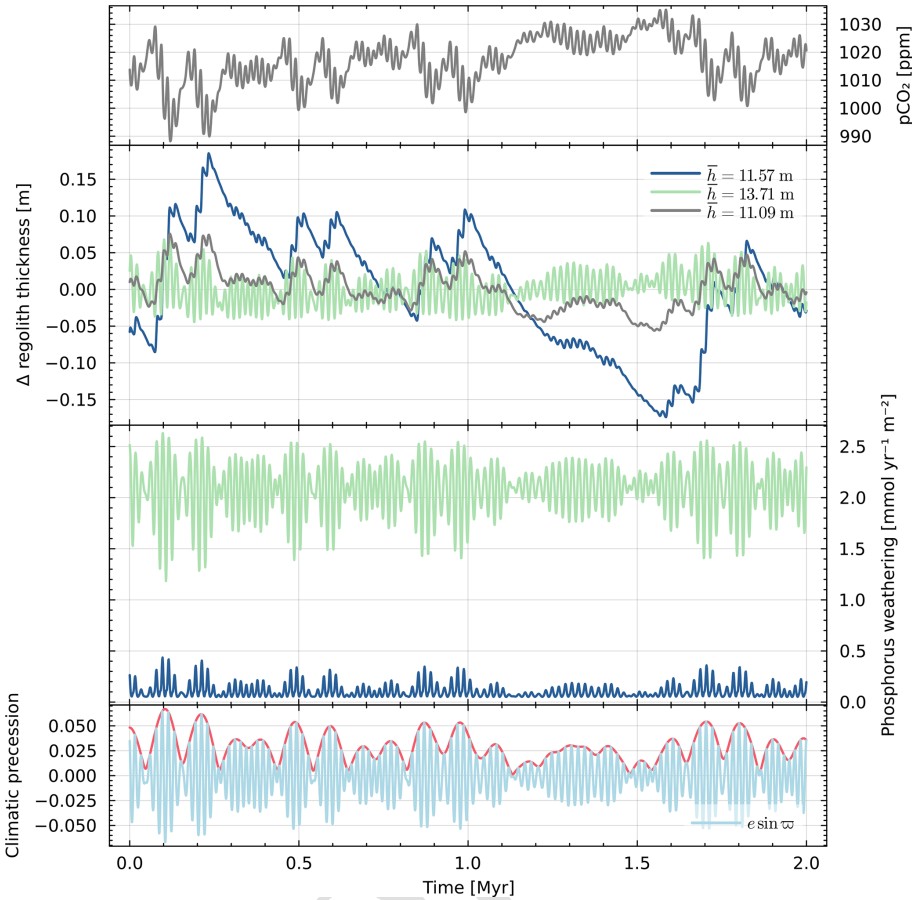

**Figure 18.** Displayed alongside the $p\mathrm{CO_2}$ and orbital forcing are the average regolith thickness variation with respect to the temporal mean $\overline{h}$ and phosphorus weathering, following the colour convention for each region defined in Fig. 16. The third grey line of the second panel depicts the global behaviour of the regolith thickness variation.

ent fluxes and anoxia operates through increased phosphorus delivery to the ocean, which stimulates primary productivity and subsequent oxygen consumption during organic matter decomposition, potentially leading to oxygen depletion. However, the small amplitude of flux variations induced by orbital forcing in our simulations is insufficient to trigger changes in oceanic dissolved oxygen concentrations which would be consistent with widespread anoxic events. While no strong conclusions regarding the precise timing and pacing of anoxic events can be drawn from these results due to the model design limitations, we have established a solid basis for exploring the dynamics of Devonian anoxia. Future work directions should include vegetation feedbacks and an improved ocean discretisation (Algeo and Scheckler, 2010; Maffre et al., 2022; Smart et al., 2023; Maffre et al., 2025).

## 5 Conclusions

In this study, we presented a dynamically coupled Gaussian Process emulator designed to bridge the gap between com-

plex General Circulation Models and geochemical simulations over geological timescales. Our approach leverages an optimised training dataset, advanced kernel functions, and robust hyperparameter tuning to capture essential climate dynamics while significantly reducing computational costs. The additive kernel framework, inspired by Duvenaud et al. (2011), allowed us to account for both individual and interacting effects of key input parameters, and the integration of PCA ensured that the emulator maintained crucial spatial teleconnections. Furthermore, the development of the emulator has taken into account the importance of a careful treatment of the Anomalous Heat Convergence flux to avoid possible global unwanted parameter-dependent forcing, and has underscored that Principal Component Analysis is effective for dimensionality reduction while a specific pre-processing of the input variables can slightly impact the emulator's performance.

The proposed method not only reproduces key features observed in the HadSM3 simulations but also provides a flexible tool for exploring the parameter space robustly and efficiently. As a case study, the setup has been applied to

a 25 Myr Devonian orbital scenario to test the relationship between orbital forcing, regolith dynamics and nutrient fluxes. Our preliminary findings indicate that high eccentricity, rather than low eccentricity, promotes regolith growth, though the variations are not significant enough to have a substantial impact on the nutrient fluxes. Therefore, it is very challenging to produce an OAE by the effect of astronomical forcing only on the atmospheric components in this setup.

By enabling the investigation of multiple astronomical scenarios at minimal computational cost, our emulator provides valuable insights into the complex relationship between orbital forcing and palaeoenvironmental changes.

Our application on the Devonian shows that we have a modelling framework stimulating research directions for understanding the mechanisms of anoxia during that period. Yet, the methodology is also applicable to other geological intervals throughout the Phanerozoic. More generally, it contributes to addressing the crucial challenge of studying Earth's climate system across a wide range of timescales in biogeochemical models, from yearly variations to million-year-long geochemical processes. We believe this work lays a solid foundation for enhancing our understanding of palaeoenvironmental variations and their driving mechanisms across the Phanerozoic.

*Code and data availability.* The code and data for training the emulator are archived on Zenodo: https://doi.org/10.5281/zenodo.15113858 (Sablon, 2025a). The vanilla version of GEOCLIM6.1.0 can be found on the public GEOCLIM GitHub repository and the modified version, including the emulator and outputs of the climate simulations presented in the article, can be found on a separated Zenodo archive: https://doi.org/10.5281/zenodo.15114774 (Sablon, 2025b). The code and data used to generate the orbital solutions are also available on Zenodo: https://doi.org/10.5281/zenodo.14894811 (Crucifix, 2025).

*Supplement.* The supplement related to this article is available online at [the link will be implemented upon publication].

*Author contributions.* PJV provided the starting dumps for the HadCM3 experiments. YG and PM provided the code of GEO-CLIM6.1.0. MC and LS conceptualized the emulator, LS wrote the code, trained the model and conducted the HadCM3, HadSM3 and GEOCLIM simulations with astronomical scenarios generated by MC. LS drafted the manuscript and all authors contributed to reviewing and editing.

*Competing interests.* The contact author has declared that none of the authors has any competing interests.

ther geographical representation in this paper. While Copernicus Publications makes every effort to include appropriate place names, the final responsibility lies with the authors. Views expressed in the text are those of the authors and do not necessarily reflect the views of the publisher.

*Acknowledgements.* We are grateful to David De Vleeschouwer (University of Münster) for his involvement in the project as scientific advisor, and specifically for his suggestion to develop plausible Devonian astronomical solutions at a meeting held in Louvain-la-Neuve in May 2023. Computational resources have been provided by the supercomputing facilities of the UCLouvain (CISM) and the Consortium des Équipements de Calcul Intensif en Fédération Wallonie Bruxelles (CÉCI).

*Financial support.* This research has been supported by the Belgian "Fonds de la Recherche Scientifique – FNRS" (Project WarmAnoxia, PDR grant no. T.0037.22).

*Review statement.* This paper was edited by Paul Halloran and reviewed by two anonymous referees.

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

**Remarks from the typesetter**

TS1    You asked us to change this expression for several instances. Since it was submitted and approved like this, we will require approval from the editor. Please prepare an explanatory file (doc or pdf) explaining the changes and why they are required.

TS2    If you want to remove this paragraph, please also write an explanation for the explanatory file for the editor.

TS3    If you want to add tildes to several ks, please also write an explanation for the editor.

TS4    Please confirm that both names are last names.

TS5    Please confirm initials.

TS6    We have updated the source here. Please confirm that the reference is displayed correctly.