# Peer review of "An Emulator-Based Modelling Framework for Studying Astronomical Controls on Ocean Anoxia with an Application to the Devonian"

_EGUsphere, 2025_

## Author Response (AR1)

September 7, 2025

**1   General comments**

We thank both reviewers for their constructive feedback and valuable suggestions that have helped improve our manuscript. Several common themes emerged from both reviews, which we address here before responding to individual comments:

**Introduction and Abstract Focus:** Both reviewers noted that the introduction reads like a science application focused on Devonian OAEs, while the abstract presents this as a model development paper. We agree that this creates a disconnect. The primary contribution of this work is methodological: we present an enhanced emulator framework that advances beyond previous approaches by implementing an additive kernel function that better captures PCA scores, accounting for ocean heat transport through a two-tier HadCM3L/HadSM3 approach, and enabling dynamic coupling with a biogeochemical model. The Devonian case study serves to demonstrate the framework's capabilities rather than being the primary focus. The abstract and introduction will be revised to highlight the methodological advances while providing sufficient context about the Devonian application and its preliminary results, including the emergent relationship between eccentricity cycles and regolith dynamics.

**HadCM3 250-year:** Both reviewers questioned the adequacy of 250-year simulations given that deep ocean equilibration in HadCM3 requires approximately 5000 years. In previous studies (e.g. in Devleeschouwer et al., 2014) the anomalous heat convergence flux is typically diagnosed by imposing a latitudinal SST curve with a vague prior on the mean value. In this study, we decided that using the coupled model would provide a more accurate representation of the orbital forcing and its spatial effects. The 250-year spin-up time, similar to Araya-Melo 2015, was chosen so that the rate of temperature change in the upper ocean layer is sufficiently low. This is a trade-off, but still represents a substantial improvement compared the previous assumption of no change in heat-convergence. It is very difficult to estimate the spatial pattern caused by not being fully at equilibrium, and we also have to consider that, given the transient nature of the astronomical forcing, pushing all experiments to equilibrium would also be an idealization. We acknowledge this limitation, and we believe that our trade-off is justified given that the focus of our study is on the continental climate response.

**Heat Convergence Fields:** As suggested by the second reviewer, the appendix will be moved to the main text. Regarding the interpolation method for the heat convergence fields, we chose Inverse Distance Weighting interpolation due to the limited number of experimental points (15 in 3 dimensions) and the lack of a priori knowledge about the underlying structure of the heat convergence response to orbital parameters. Leave-one-out cross-validation across the 12 monthly fields yields an average correlation coefficient of 0.7. While this performance is not perfect, it is

substantially better than a nearest neighbor approach, which would have effectively partitioned the parameter space into discrete regions rather than providing smooth interpolation between experimental points.

**Figure Improvements:** We have revised several figures to use geographic map projections instead of the previous cartesian `lon-lat` projection, which improves the visibility and interpretability of spatial patterns.

Below we address the remaining individual comments from each reviewer.

**2  Response to Reviewer 1**

**Major Comments**

> **Comment 1**
>
> Section 3.7 contains an insightful overview of the emulator behaviour and describes how pCO2 and the three orbital parameters influence the surface temperature. What I am missing is a similar evaluation for the runoff emulator. Emulator behaviour is key in this study. To understand the response of runoff and P weathering to orbital forcing, we need to know its primary control. A figure a la Fig.6 for runoff would help. Because understanding spatial patterns becomes important, I'd also suggest including spatial maps (in supplement?) that compare the temperature and runoff between different orbital states (e.g. high vs low obliquity, high vs low precession/eccentricity). I am aware of the already large number of figures, but I do think these improve interpretation beyond the PCA maps, especially to support the differing response to precession in different regions that is not obvious in Fig.6.

We thank the reviewer for this suggestion. We agree that a comprehensive analysis of runoff emulator behavior parallel to the temperature analysis in Section 3.7 (Figures 11-12) would significantly enhance the manuscript. As the reviewer correctly notes, understanding runoff controls is crucial since runoff is the primary driver of weathering and erosion in GEOCLIM. We will prepare supplementary figures showing: (1) spatial visualization of the runoff principal components and (2) spatial maps comparing temperature and runoff between contrasting orbital states (high vs. low obliquity, high vs. low precession/eccentricity). To avoid overloading the main text, these will be added as supplementary material. See Supplementary Figures 1 to 8.

> **Comment 2**
>
> The main finding is that the regolith stack grows during high eccentricity but the impact on nutrient fluxes is small. After reading the discussion a few times, it is still unclear to me how regolith dynamics and nutrient fluxes become so decoupled. It is surprising that the regolith 'functions as a nutrient reservoir' (L.515) but when the regolith shrinks (blue Fig.15), P fluxes are barely affected. Can you explain what the P weathering flux represents, e.g. the flux to the ocean, or the P flux from bedrock to regolith, something else? To help the interpretation, it might be useful to discuss or plot how the total P inventory in the regolith changes over time, rather than the weathering fluxes, which will also aid to reconcile global vs regional effects. Also clarifying the link between regolith thickness, erosion, weathering, and nutrient fluxes is necessary. Is regolith thickness the only control on weathering? How does P move between the bedrock, regolith, and ocean? These questions call for a better explanation of their representation in GEOCLIM, either in the method section (add to 2.1.2) or at the start of Section 4. Because it is such a critical component of this study, it is worth explaining as opposed to referring the reader to a supplement in a different paper. The conclusion that high eccentricity results in a thicker regolith stack on does make sense on L.503-511 (although see point 5 below) and also explains the 2.4 Ma node, but the decoupled nutrient flux remains unclear.

We agree with the reviewer that this subject needs more explanation. The apparent decoupling between regolith thickness and phosphorus fluxes reflects the specific formulation in GEOCLIM6.1.0. In this model, phosphorus weathering is not directly proportional to regolith thickness but rather comes from three sources: silicate weathering (the dominant contributor), carbonate weathering, and petrogenic organic carbon weathering. Of these three sources, only silicate weathering is dynamically computed in Dynsoil and therefore affected by regolith thickness.

To clarify these dynamics, we will add a more detailed explanation of the regolith-weathering coupling in Section 2.1.2. The term *reservoir* is misleading as there is no discrete P inventory and will be removed from the revised manuscript.

**Minor Comments**

> **Comment 1**
>
> Methods - emulator: I am happy to see a detailed description of the methods to build and rigorously test the emulator. This provides full transparency and allows reproducibility. Again, I am not able to comment in detail on the maths or statistical choices, but I found the broad lines of Section 3 relatively comprehensible despite the complexity. I do have one suggestion. Because of the many components and length of Section 3, it would be useful to include a visual flow chart of the methods or a numbered list of the steps taken before describing the details. This gives the reader an initial summary overview of what follows and puts each section in the broader methodological context.

We agree that Section 3 would benefit from a methodological overview. We will add a visual flowchart at the beginning of Section 3 summarizing the key steps: (1) Design of parameter space and Latin Hypercube Sampling, (2) GCM simulations with HadCM3L/HadSM3, (3) PCA decomposition of climate fields, (4) GP regression with additive kernel for each PC, (5) Hyperparameter

optimization, (6) Validation, and ultimately (7) Integration into GEOCLIM.

> **Comment 2**
> Section 4.2. The results of Fig.10 depend on the dynamic pCO2; I'd like to know how CO2 varies in this simulation in terms of amplitude in addition to its periodicity shown in Fig.11 to get a sense of the relative contribution of CO2 vs orbits that influence the temperature and runoff power spectra. Or is it comparable to the CO2 shown in Fig.14 where obliquity is constant? Further, describe the primary mechanism(s) that control pCO2 in your astronomical GEOCLIM simulations.

It is indeed comparable to Fig. 14; a mention of this will be added to the revised manuscript. Concerning the mechanisms controlling pCO2, these are the negative feedbacks implemented in the GEOCLIM model, to which the reader is referred for description. In this case, the negative feedback loop defined in Walker et al. (1981) is the main mechanism.

> **Comment 3**
> Section 4.3. The emulator predicts temperature and runoff. These are annual mean fields–clarify this early in the method section along with a mention of the GEOCLIM model timestep. Can you address whether you expect the lack of seasons to affect the results? I understand GEOCLIM cannot simulate seasons, but this should be mentioned for completeness and the implications discussed. For instance, it is reasonable to expect that regions experiencing intense seasons might have different regolith accumulation than regions with milder seasons, despite the fact these could have the same annual mean temperature or runoff. An advantage of the hypothesis tested here is that it primarily affects the tropics so perhaps seasons don't play a big role but do acknowledge this regardless in case the emulator is used to test other hypotheses in the future.

GEOCLIM is not designed to simulate seasonal variability; this is discussed in the GEOCLIM7 description paper, to which we refer the reader for more details. In theory, however, the emulator could be adapted to account for seasonal effects by emulating monthly mean climate and linearly interpolating through time. This approach would, however, require a significant decrease in the model timestep, leading to a large increase in computational cost. It should be noted that the seasonal cycle is captured in HadCM3, so that the orbital forcing affecting the seasonal cycle effectively affects the annual mean fields. This is how, for example, monsoons respond to precession.

> **Comment 4**
> Section 4.4. The impact of the small P fluxes on ocean oxygen concentrations are not shown because they are small. This seems reasonable though the title of the manuscript suggests that you will do otherwise. Perhaps a more targeted title is appropriate. In light of this, the link between nutrient fluxes and anoxia is not evidently explained in the text. Add a brief sentence describing this link in the context of GEOCLIM.

In the context of GEOCLIM, and particularly the version used in our study, it is impossible to create an OAE from orbital forcing alone. Section 4.4 will be modified to clarify the limitations of our approach and the specific role of nutrient fluxes in the context of GEOCLIM. Some time series of oxygen concentrations will potentially be added to the supplementary material to support the discussion.

**Brief Comments and Typos**

> **Comment 5**
>
> - Title. Should it be: "An Emulator-Based Modelling Framework for Studying Astronomical Controls on Ocean Anoxia with an Application  to the Devonian"?
>
> - Figures throughout: some subpanels are labelled (a), (b), etc... others aren't. Keep consistent throughout.
>
> - Line 11-13. Consider changing sentence to the following for clarity: "These challenges can stem from uncertainties in boundary conditions, forcing mechanisms, and geological reconstructions that significantly complicate model parameters tuning, or from uncertainties in the timescales involved."
>
> - Line 24. "..would help trigger regional or  possibly global anoxia  when low-oxygen concentrations prevail."
>
> - Line 45. '...calibrated on a  of experiments...'. Use 'suite' or 'ensemble' instead of plan
>
> - Line 104-105. Also reference Fig.2
>
> - Line 159. Define GP emulator first time it's mentioned
>
> - Line 359. "When such outliers occur, they deserve..."
>
> - Line 440-447. Clarify if the calculation of temperature in ocean boxes is done every 25 yr timestep, or just once at the start as for water mass exchange.
>
> - L.461-462. Make the link between regolith, nutrients, and ocean oxygen.
>
> - L. 485. Explain '.. long-term response in the subsystem'. This implies some memory effect or nonlinearity.
>
> - Figure 10. The colours in panels c and d are difficult to see. Consider using a different colormap or change the background from white to a darker colour so the light lines become visible. Or change the scaling of the colorbar?

We thank the reviewer for spotting the typos and will apply all of those corrections.

**3    Response to Reviewer 2**

**Major Comments**

> **Comment 1**
> The abstract has more of the above approach, presenting a model development paper, though this could be developed in similar direction to the above ideas. In the final sentence, and opposite to above, I would close with something more explicit about the science case study and results, however inconclusive, perhaps the emergent relationship between high eccentricity and soil development. It's a shame to leave this completely vague.

As stated in Section 1 of this document, the abstract and introduction will be revised to better align with the journal's focus. Your suggestion will be taken into account while revising these sections.

> **Comment 2**
> At the end of all this work there is a product, the coupled model. Can this be provided in a useable form? Does the model need a name and a version number as per the GMD convention?

We will make the complete coupled model code and trained emulator available in a useable form through two public repositories. We designate the updated version of GEOCLIM6.1 as `GEOCLIM6.1-EMUL`.

> **Comment 3**
> Section 2.1.2. The study uses GEOCLIM, which the authors describe as a 10-box atmosphere-ocean model coupled to a continental model that is spatially resolved. Could more detail be on the continental model be provided, as it is here where the coupling with the emulator is most relevant? Is it for instance configured to the same (Fig 1) topography as HadCM3 simulations, at the same resolution? Also, the ocean model – do the nine boxes represent depth and/or spatial resolution? A figure representing the ocean and continental models might help, perhaps in the Appendix.

We will expand Section 2.1.2 to provide more detail on the continental model setup. The continental module uses the same Late Devonian paleogeography (370 Ma) as the HadCM3/HadSM3 simulations, ensuring consistency between the climate emulator and biogeochemical model. The nine ocean boxes represent the following discretization: polar deep and surface (including thermocline) for both northern and southern hemispheres, mid-latitude deep, intermediate (thermocline), and surface, and epicontinental deep and surface. While a complete description exists in previous publications, we will add a concise summary of the key elements and clarify the box geometry without adding a full schematic to maintain focus on the emulator coupling methodology.

**Minor Comments**

> **Comment 1**
> Line 155 obliquity is varied between 20.75° and 23.75° for the HadSM3 ensemble. Please discuss why this specific range was chosen (where do the limits come from?). Related, in the HadCM3 ensemble, the range is between 21° and 25°, so worth noting the slight extrapolation implied. Was there a motive for this choice in the HadCM3 ensemble, presumably it was designed with more general applicability in mind?

The state of precession and obliquity during the Devonian is an ongoing topic. It depends on different factors including the Earth-moon distance and dynamical ellipticity. When we first designed the more expensive HadCM3 experiments, we adopted conservative ranges centered on modern values. A more careful review (including works by Farhat 2022) led us to adjust our experimental range for the HadSM3 experiments.

> **Comment 2**
> Section 3.1, here I would argue that a shorter summary of GP would be adequate (perhaps just lines 197-203 and 230-238?), and much of this detail would be better in an Appendix. GP has been around a long time and is widely applied. For my own interest I checked and see Rasmussen 2006 has 39,000 citations and Mackay 1998 has 17,000.

We agree that Section 3.1 could be streamlined. Given the extensive literature on Gaussian Processes, we could move some details to an appendix. However, since the choice of a less common kernel function is a fundamental element of our study, we prefer keeping this information as part of the main text.

> **Comment 3**
> Line 241. Should probably cite Holden and Edwards 2010 which first presented dimensionally reduced emulation for climate to advance beyond pattern scaling.
> https://agupubs.onlinelibrary.wiley.com/doi/full/10.1029/2010GL045137

We thank the reviewer for this suggestion and will add this citation to acknowledge the pioneering work on dimensionally reduced climate emulation, along with the Wilkinson et al. 2010 paper which appeared the same year.

> **Comment 4**
> 260 please quantify "unsatisfactory" – this is subjective, and quantification would enable the reader to evaluate the cost-benefits of your careful kernel choice. OK, I see this quantification is done later – would be clearer to express this differently, perhaps simply point to the analysis coming later. Or maybe just say you considered two approaches and not yet mention which was preferred.

We will revise line 260 to state something close: "We evaluated two kernel approaches and found performance differences that are quantified in Section 3.6" rather than using the subjective term "unsatisfactory" as per your suggestion.

> **Comment 5**
> Section 3.4. Can any of this be moved to the appendix, for instance 311-320 describing a more complex objective function which was not used as too computationally expensive. There is no quantification, either of the value added or the computational speed, and including this in the main body of text adds little value I can see.

We agree and will move the discussion of the alternative objective function (negative expected log pointwise predictive density) to an appendix, keeping only a brief mention in the main text that alternative optimization approaches were considered.

> **Comment 6**
> Line 382 were seven PCs used, or six as plotted in Figs 6 and 7?

We will clarify this discrepancy - seven PCs were used for the emulation as stated in the validation section, but only the first six are shown in the figures for clarity.

> **Comment 7**
> Figs 6, 7. You plot Sobol indices. A definition and/or citation would be useful.

A citation referring to https://doi.org/10.1016/S0378-4754(00)00270-6 will be added.

> **Comment 8**
> Line 440. Can you better explain/justify this. How do you map spatial fields of emulated surface air temperature onto global average SST. Moreover, why do you use a fixed linear regression using outputs from a different model, which presumably loses the spatial relationships between SST and orbit. Why not relate your patterns of emulated SAT directly onto patterns of SST needed by GEOCLIM?

This is an important limitation of our current approach that uses a slab model and hence lacks vertical resolution in the ocean. However, one needs to specify the temperature of each box. With GEOCLIM, two options are possible: either use a parameterized CO2-Temperature relationship or use GCM outputs. Neither of these options is satisfactory, as the first does not capture astronomical forcing (or not at all in some cases) and the second suffers from the choice of a slab model. To deal with this issue, we decided to use a hybrid approach: parameterize the box temperature as a function of the global mean sea surface temperature. The parameters are fitted using outputs from a coupled model of intermediate complexity (namely cGENIE), but ultimately in the GEOCLIM model, the emulated global mean SST is considered and hence takes into account both orbital forcing and pCO2 fluctuations.

> **Comment 9**
> Line 465. You generate a plausible yet hypothetical forcing scenario. I wonder if this explains your apparent reticence with conclusions. The coupled model is presumably very fast, though I missed whether that was quantified anywhere. Could you not e.g. generate an ensemble of forcing input time series to sample plausible scenario space and use this to generate a distribution of outputs that could be summarised into something more robust? (Not necessary in the context of a model development paper, but posing the question.)

Regarding speed, coupling GEOCLIM to the emulator comes with no extra cost at all, as it relies on well-optimized `BLAS` routines and most computational time is spent in other routines.

The second part is an excellent suggestion, and another study now using the latest GEOCLIM7 and its ability to easily modify ocean resolution is ongoing and will explore this aspect in more detail.

**Supplementary Figures**

[Figure]

Figure 1: The figure presents the first six principal components of the surface temperature, ordered by their eigenvalues and rescaled by a multiplicative factor. The colour scale is indicative of the amplitude and its sign within a single PC, but is independent across different PCs. The black lines delineate the coastlines of the continental configuration.

[Figure]

Figure 2: The figure presents the first six principal components of the surface runoff, ordered by their eigenvalues and rescaled by a multiplicative factor. The colour scale is indicative of the amplitude and its sign within a single PC, but is independent across different PCs. The black lines delineate the coastlines of the continental configuration.

[Figure]

Figure 3: Comparison of surface runoff for different climatic precession values. The top two panels depict extremal cases, where $\sin\varpi$ equals 1 for the left one and $-1$ for the right one. The bottom panel represents the difference of those two states. $p\mathrm{CO}_2$ is constant at 1000 ppm.

[Figure]

Figure 4: Comparison of surface temperature for different climatic precession values. The top two panels depict extremal cases, where $\sin\varpi$ equals 1 for the left one and $-1$ for the right one. The bottom panel represents the difference of those two states. $p\mathrm{CO}_2$ is constant at 1000 ppm.

[Figure]

Figure 5: Comparison of surface runoff for different climatic co-precession values. The top two panels depict extremal cases, where $\cos\varpi$ equals 1 for the left one and $-1$ for the right one. The bottom panel represents the difference of those two states. $pCO_2$ is constant at 1000 ppm.

[Figure]

Figure 6: Comparison of surface temperature for different climatic co-precession values. The top two panels depict extremal cases, where $\cos\varpi$ equals 1 for the left one and $-1$ for the right one. The bottom panel represents the difference of those two states. $pCO_2$ is constant at 1000 ppm.

[Figure]

Figure 7: Comparison of surface runoff for different obliquity values. The top two panels depict extremal cases, where the obliquity is low for the left one and high for the right one. The bottom panel represents the difference of those two states. $p\mathrm{CO}_2$ is constant at 1000 ppm.

[Figure]

Figure 8: Comparison of surface temperature for different obliquity values. The top two panels depict extremal cases, where the obliquity is low for the left one and high for the right one. The bottom panel represents the difference of those two states. $p\mathrm{CO}_2$ is constant at 1000 ppm.